# Efficient LLM Moderation with Multi-Layer Latent Prototypes

**Maciej Chrabąszcz** [1 2]  **Filip Szatkowski** [2 3]  **Bartosz Wójcik** [4]
**Jan Dubiński** [1 2]  **Tomasz Trzciński** [2 3 5]  **Sebastian Cygert** [1 6]

## Abstract

Although modern LLMs are aligned with human values during post-training, robust moderation remains essential to prevent harmful outputs at deployment time. Existing approaches suffer from performance-efficiency trade-offs and are difficult to customize to user-specific requirements. Motivated by this gap, we introduce Multi-Layer Prototype Moderator (MLPM), a lightweight and highly customizable input moderation tool. We propose leveraging prototypes of intermediate representations across multiple layers to improve moderation quality while maintaining high efficiency. By design, our method adds negligible overhead to the generation pipeline and can be seamlessly applied to any model. MLPM achieves state-of-the-art performance on diverse moderation benchmarks and demonstrates strong scalability across model families of various sizes. Moreover, we show that it integrates smoothly into end-to-end moderation pipelines and further improves response safety when combined with output moderation techniques. Overall, our work provides a practical and adaptable solution for safe, robust, and efficient LLM deployment.

## 1. Introduction

Large language models (LLMs) have quickly become central to modern applications, making their safety and alignment with human values increasingly important. While techniques like RLHF (Bai et al., 2022; Ouyang et al., 2022) and instruction tuning (Li et al., 2025) have substantially improved model safety, even state-of-the-art models remain susceptible to emergent risks even for aligned models (Andriushchenko et al., 2025; Carlini et al., 2023; Liu et al., 2023). As a result, ensuring practical LLM safety requires an additional evaluation of the model inputs and responses. This has motivated recent advances in moderation tools (Lee et al., 2025a; Zheng et al., 2024), which are considered an essential component of safe deployment of the LLMs.

The most common moderation tools can be broadly categorized as either specialized guard models or latent-based methods. In practice, the choice between these approaches typically reflects preferences over training cost, performance, and efficiency. Guard models (Dong et al., 2024; Ghosh et al., 2024; Han et al., 2024; Inan et al., 2023; Sharma et al., 2025) offer good performance, but introduce additional model into the moderation pipeline, complicating the deployment and increasing its cost. Training guards is resource-heavy and requires carefully curated datasets, limiting most users to a fixed set of pre-trained models. On the other hand, latent-based methods (Ayub & Majumdar, 2024; Abdelnabi et al., 2025) are usually lightweight, but offer worse performance than guard models. None of these approaches fully satisfies the combined requirements of performance, efficiency, and adaptability to custom safety policies, which are essential for real-world deployment.

We address the above-mentioned gap with Multi-Layer Prototype Moderator (MLPM), a lightweight, latent-based input moderation approach that uniquely combines the efficiency and flexibility inherent to latent-based methods with state-of-the-art performance exceeding that of the best guard models. We provide a high-level comparison of the existing methods and our approach in Table 1. MLPM uses the internal states of off-the-shelf LLMs to assess input safety via distance to safe and unsafe prototypes. We achieve superior performance by unifying the Mahalanobis distance-based classifier (Goswami et al., 2023) with a multi-layer prototype strategy. This approach allows us to harness the diverse semantic information distributed across intermediate layers (Masarczyk et al., 2023; Szatkowski et al., 2025; Zou et al., 2023a) for more accurate and robust detection. This holistic view enables the correct classification of inputs that appear ambiguous when observing any single layer in isolation. MLPM delivers guard-level performance, is model-

---

[1]NASK National Research Institute, Warsaw, Poland [2]Warsaw University of Technology, Warsaw, Poland [3]IDEAS Research Institute, Warsaw, Poland [4]Jagiellonian University, Cracow, Poland [5]Tooploox [6]Gdańsk University of Technology, Gdańsk, Poland. Correspondence to: Maciej Chrabąszcz <maciej.chrabaszcz@nask.pl>, Filip Szatkowski <filip.szatkowski.dokt@pw.edu.pl>.

*Proceedings of the 43rd International Conference on Machine Learning*, Seoul, South Korea. PMLR 306, 2026. Copyright 2026 by the author(s).

*Table 1.* High-level conceptual comparison of existing input moderation approaches and our method. MLPM is able to uniquely combine low training cost, efficient inference, and flexibility inherent to latent-based approaches with state-of-the-art moderation performance.

| | Training Cost | Data-Efficiency | Inference Efficiency | Memory Footprint | Flexibility | Safety Assessment Performance |
|---|---|---|---|---|---|---|
| Guard Models | High | Low | Low | High | Low | High |
| Latent-based methods | Low | High | High | Low | High | Medium |
| **MLPM (ours)** | **Low** | **Very High** | **High** | **Low** | **High** | **High** |

agnostic, and adds minimal compute and memory overhead at inference. Furthermore, it can be trained cheaply and efficiently, even in data-constrained scenarios, achieving guard-level performance with as few as 1,000 samples.

We evaluate MLPM across diverse input moderation benchmarks and demonstrate that it achieves state-of-the-art performance, outperforming the alternative guard and latent-based approaches across various model families and sizes. We further demonstrate how MLPM can be easily integrated into end-to-end moderation pipelines alongside output moderation tools, enhancing the overall safety of LLM systems. Finally, through detailed ablations, we investigate the robustness of our method in low-data regimes, out-of-distribution scenarios, and the intriguing multi-layer dynamics underlying its performance. Our key contributions are:

- We introduce MLPM, an efficient LLM input moderation approach that uses Mahalanobis distance across multi-layer representations to assess prompt safety.

- We demonstrate that MLPM achieves state-of-the-art performance, surpassing existing latent-based methods and Guard models across diverse benchmarks.

- We show how MLPM seamlessly integrates as a conditioning signal for steering methods, reducing unwanted refusals and enhancing end-to-end deployment safety.

Taken together, our work provides a state-of-the-art, efficient, and customizable solution for LLM moderation.

## 2. Related Work

**LLM alignment and safety.** LLMs are usually pretrained on large corpora of data that are impossible to fully supervise. Therefore, pre-training is typically followed by supervised finetuning that ensures the alignment of the model with human preferences and values (Bai et al., 2022; Dai et al., 2024; Li et al., 2025; Lim et al., 2025; Ouyang et al., 2022). However, various studies prove that even the most popular frontier models are still prone to generating unsafe responses (Carlini et al., 2023; Liu et al., 2023; Zou et al., 2023b), as safety alignment can be superficial and lose its effect after the initial few tokens of generation (Qi

et al., 2025). Furthermore, focusing strictly on the safety alignment might negatively affect model capabilities (Huang et al., 2025; Wei et al., 2023; Wolf et al., 2024).

**LLM moderation.** Moderation techniques ensure LLMs comply with established guidelines while preserving their capabilities and output quality. Generally, moderation can be applied either to the model output during text generation or to the input requests before the generation starts. Output moderation assesses the already generated LLM responses, and typically relies on guard models (Inan et al., 2023; Han et al., 2024; Ghosh et al., 2024; Sharma et al., 2025; Yin et al., 2025), rule-based approaches (Clarke et al., 2023; Kumar et al., 2024), prompt engineering (Zheng et al., 2024; Xie et al., 2023), activation steering (Zou et al., 2023a; Luo et al., 2024; Lee et al., 2025a; Qiu et al., 2024), or specialized fine-tuning (Zou et al., 2024; Zhang et al., 2024). While effective at ensuring the safety of responses, steering techniques can often hinder the model's capabilities (Lee et al., 2025a). Refusing to engage with malicious prompts is often a sufficient and efficient approach (Manczak et al., 2024), which motivated the development of input moderation strategies that evaluate prompts before generation. Common input moderation techniques employ guard models or latent-based methods that leverage model representations to detect malicious content (Abdelnabi et al., 2025; Ayub & Majumdar, 2024). Guard models offer high performance at a high cost, while latent methods are lightweight but less effective. Our method presents a novel latent-based approach to input moderation, combining prototype-based classification with multi-layer feature aggregation to achieve guard-level state-of-the-art performance, while remaining efficient and easily adaptable to specific safety policies.

## 3. Multi-Layer Prototype Moderator

The goal of input moderation is to assess whether the input prompt $x$ belongs to the class of harmful prompts $\mathcal{X}_{\text{harm}}$, which can lead the LLM to produce an unsafe response. This allows the generation process to be halted, ensuring safe interaction and avoiding unnecessary computation. In this section, we present Multi-Layer Prototype Moderator (MLPM), a latent-based input moderation approach that combines high-tier performance and efficiency. By utiliz-

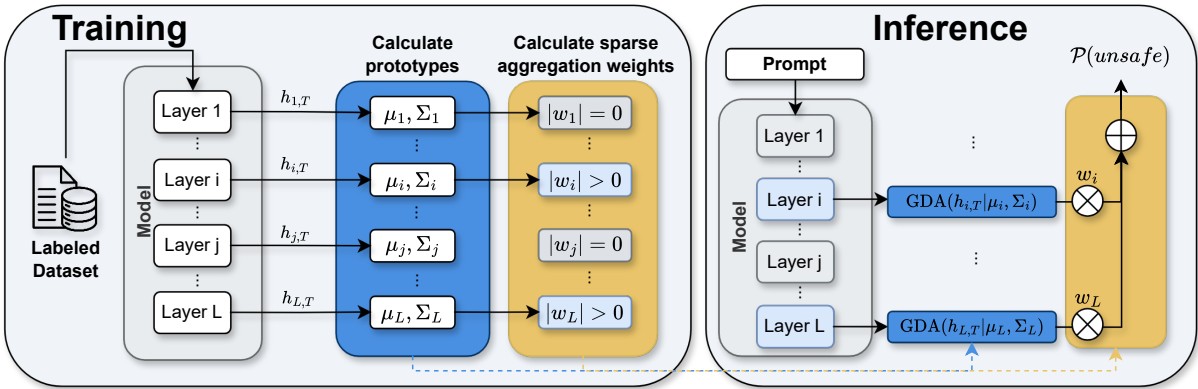

*Figure 1.* Our proposed Multi-Layer Prototype Moderator (MLPM) framework. **During training**, we compute class-conditional prototypes $(\mu_i, \Sigma_i)$ based on last-token hidden representations $(h_{i,T})$, which we use to define a per-layer Gaussian Discriminant Analysis (GDA) classifier. We then learn sparse aggregation weights $(w_i)$ over the GDA scores. **During inference**, we use the pre-trained GDA classifiers to compute classification scores from layers with non-zero weights $(|w_i| > 0)$, and produce a safety probability, $P(\text{unsafe})$, from their weighted aggregate. MLPM enables state-of-the-art performance with lightweight training and negligible inference overhead.

ing multi-layer representations and prototype classification, MLPM provides state-of-the-art safety assessment for any model of the user's choice. At the same time, our design ensures that our method remains flexible and easy to deploy, even when computational resources and data are scarce. An overview of our method is shown in Figure 1.

### 3.1. Prototype-based classification

Nearest Mean Classifier (NMC) is widely used in deep learning due to its interpretable decision boundaries, strong generalization, and data-efficiency (Wang et al., 2020b; Xian et al., 2017; Rebuffi et al., 2017; Łapacz et al., 2025). However, despite its effectiveness, prior to our work, NMC has not previously been applied to LLM safety. NMC assigns a class $c$ to the new sample $x$ based on the distance between the representations of $x$ and the prototypes of all the classes embedded in the latent space. The latent representation is obtained via feature extractor $g$, which maps the input $x$ to a latent vector $h = g(x)$. In most cases, the extractor corresponds to a part of the neural network, though in principle it can be any function that produces useful embeddings. The prototypes of all classes are computed as the empirical means of their corresponding samples in the latent space:

$$\mu_c = \frac{1}{N_c} \sum_{i=1}^{N_c} h_i^{(c)}, \qquad (1)$$

where $h_i^{(c)} = g(x_i^{(c)})$ is the latent representation of the $i$-th training example in class $c$, and $N_c$ is the number of examples in that class.

The simplest NMC determines the class of $x$ by comparing

the distances between $h = g(x)$ and the class prototypes:

$$c = \operatorname*{arg\,min}_{j \in \{1, \ldots, C\}} \mathrm{d}\left(h, \mu_j\right), \qquad (2)$$

where $C$ refers to the number of considered classes.

The performance of NMC depends mainly on the choice of the feature extractor and the distance metric. While Euclidean distance is the most straightforward option, numerous variants of NMC have explored alternative metrics to improve performance. In particular, Mahalanobis distance has been shown to generalize better and provide increased robustness (Wang et al., 2020a; Goswami et al., 2023; Wang et al., 2024). By accounting for data variance, the Mahalanobis distance captures the underlying geometry of the latent space, enabling more accurate discrimination between clusters with different shapes and orientations (Lee et al., 2018). Motivated by the mentioned properties, we adopt Mahalanobis distance in our method, as it is better suited to capture the complex structure of LLM representations[1].

Mahalanobis distance between a vector $h \in \mathbb{R}^d$ and the prototype represented by mean $\mu_c$ also accounts for the covariance matrix $\Sigma_c$, and is defined as:

$$d_M(h, \mu_c, \Sigma_c) = \sqrt{(h - \mu_c)^\top \Sigma_c^{-1}(h - \mu_c)}, \quad (3)$$

where we estimate $\Sigma_c^{-1}$ with a Bayes ridge-type estimator (Kubokawa & Srivastava, 2008).

With the choice of Mahalanobis distance, NMC can be altered to estimate class probabilities via Gaussian Discriminant Analysis (GDA), where the probability that an input $x$

---

[1]Our intuition is further supported empirically in Table 4

with latent representation $h = g(x)$ belongs to class $c$ is:

$$\mathcal{P}(c|x, \mu_c, \Sigma_c) = \frac{\exp(d_M(h, \mu_c, \Sigma_c))}{\sum_{i=1}^{k} \exp\left(d_M(h, \mu_i, \Sigma_i)\right)}. \quad (4)$$

For safety assessment in our setting, we classify inputs between two classes, $\mathcal{X}_{\text{safe}}$ and $\mathcal{X}_{\text{harm}}$. We use the LLM as our feature extractor, and we only use the representation corresponding to the last token in the prompt. Importantly, at inference time, we leverage the hidden states already computed by the model during the prefill stage, so the only additional overhead introduced by MLPM is the negligible cost of performing GDA. To further reduce memory requirements, we use a shared covariance matrix $\Sigma_c = \Sigma$.

## 3.2. Combining multi-layer prototypes

As discussed in the previous sections, representations from different layers capture diverse information that can be informative for safety assessment. Motivated by this observation, MLPM leverages intermediate representations to improve moderation performance. To this end, we apply the GDA procedure described in Section 3.1 to a selected subset of intermediate layers and perform a weighted aggregation of the resulting predictions from the selected subset of layers.

Specifically, to construct MLPM classifier, we extract the final token representations at the outputs of the feed-forward networks (FFNs) from each of the $L$ transformer blocks in the LLM. For each layer $l$, we compute class-conditional prototypes parameterized by the mean $\mu_c^l$ and inverse of shared covariance matrix $(\Sigma^l)^{-1}$. Substituting the layer-specific representations for $h$ in Equation (4), we obtain layer-wise GDA classifiers that calculate probability $\mathcal{P}^l(x \in \mathcal{X}_{\text{harm}}|x, \mu^l, \Sigma^l)$ of the input sample $x$ being harmful.

Since intermediate-layer representations differ in their relevance to safety assessment and may encode overlapping information, we aggregate layer-wise GDA predictions using learned weights that select informative layers and control their influence on the final prediction. Formally, the probability assigned by MLPM to a sample $x$ is computed as a weighted aggregation of layer-wise GDA predictions:

$$\text{MLPM}(x) = \sigma\Big(\sum_{l=1}^{L} w^l \cdot \mathcal{P}^l(x \in \mathcal{X}_{\text{harm}}|x, \mu^l, \Sigma^l)\Big), \quad (5)$$

where $\mathbf{w} = w^1, \ldots, w^L$ are aggregation weights and $\sigma$ is a sigmoid function. We learn these weights with an $\ell_1$ regularization penalty on $w$. We optimize this objective over the entire training dataset, with the regularization strength controlled by the hyperparameter $C$. This penalty promotes sparsity and robustness in the aggregation, mitigating redundancy among similar layers and avoiding unnecessary computation. Consequently, MLPM is able to leverage informative signals from distinct representations, thereby improving overall robustness and performance.

## 3.3. Practical implications of MLPM design

MLPM is designed to maximize the safety assessment performance, and at the same time minimize both computational and memory overhead. Due to its low training cost and high data efficiency, our method can be applied to any LLM of choice, does not require substantial resources, and only slightly increases the complexity of model inference, providing a lightweight, flexible and self-contained solution.

### 3.3.1. TRAINING EFFICIENCY

The training process of MLPM is notably lightweight, as it uses the representation of single, last-token, and requires only a single forward pass through the prompt dataset, without gradient calculations or text generation. The prototype calculation and computation of aggregation weights can run quickly even on a CPU. As shown in Section 4.3, MLPM is also remarkably data-efficient and can perform safety assessment effectively even when trained on a small dataset.

### 3.3.2. INFERENCE EFFICIENCY

At the inference time, the overhead of MLPM is negligible, as the intermediate representations are already computed when prefilling the prompt. We can estimate the computational overhead of MLPM during inference by analyzing the ratio of FLOPs required for safety assessment with our method relative to the total prefill FLOPs; in the worst cases, this overhead amounts to less than $0.001\%$ of the prefill compute. In addition to its low computational cost, MLPM is highly memory-efficient, using just $\sim 24\text{KB}$ per prototype stored in half-precision for the Llama3.1-8B model. Ultimately, the combination of negligible computational and memory inference overhead establishes MLPM as a highly practical solution for safety enforcement during LLM deployment. See Appendix L for the details on our estimation.

## 4. Experiments

In this section, we provide a detailed comparison of MLPM with alternative input moderation approaches, focusing on the most relevant guard models and latent-based methods. Among the guard models, we use Aegis-Defensive (Ghosh et al., 2024), LlamaGuard3 (Inan et al., 2023), Granite Guardian (Padhi et al., 2024), ShieldGemma (Zeng et al., 2024), and WildGuard (Han et al., 2024). We also use latent-based approaches such as Abdelnabi et al. (2025) and Ayub & Majumdar (2024). As base models for latent-based methods (including MLPM) we use Mistral (Jiang et al., 2023), Llama (Grattafiori et al., 2024), OLMo (OLMo et al., 2024), and Qwen3 (Yang et al., 2025), which allows us to assess our approach across diverse models.

We evaluate the methods on 8 prompt harmfulness datasets, including WildJailbreak (Jiang et al., 2024) and WildGuard-

*Table 2.* F1 score on harmful datasets. MLPM consistently outperforms prior latent-based approaches when applied to the same model. Additionally, MLPM surpasses resource-heavy guard models, including LlamaGuard 3, ShieldGemma, and GraniteGuardian. Notably, when applied to OLMo2, MLPM achieves the highest overall performance, outperforming even the strongest guard baseline, WildGuard.

| Dataset | Aegis | HarmB | OpenAI | SimpST | TChat | WGMix | WJB | XSTest | Average |
|---|---|---|---|---|---|---|---|---|---|
| **Latent-Based** | | | | | | | | | |
| Llama-8B-Inst+Ayub & Majumdar (2024) | 82.52 | 96.98 | 66.60 | 98.48 | 55.62 | 80.91 | 82.85 | 92.76 | 82.09 |
| Llama-8B-Inst+Abdelnabi et al. (2025) | 84.16 | 95.18 | 67.99 | 98.99 | 59.59 | 86.27 | 93.27 | 90.63 | 84.51 |
| Llama-8B-Inst+**MLPM(Ours)** | 85.13 | 99.58 | 72.85 | 99.50 | 69.17 | 88.04 | 94.69 | **97.44** | 88.30 |
| Qwen3-8B-Inst+Ayub & Majumdar (2024) | 80.00 | 90.62 | 74.56 | 95.29 | 68.90 | 80.64 | 81.31 | 90.21 | 82.69 |
| Qwen3-8B-Inst+Abdelnabi et al. (2025) | 84.47 | 99.37 | 68.45 | 97.96 | 60.94 | 85.18 | 90.44 | 88.06 | 84.36 |
| Qwen3-8B-Inst+**MLPM(Ours)** | 83.49 | **100.0** | 72.35 | 96.91 | 64.40 | 86.21 | 92.00 | 92.23 | 85.95 |
| Mistral-7B-Inst+Ayub & Majumdar (2024) | 79.60 | 90.87 | 75.69 | 98.99 | 63.44 | 83.31 | 87.53 | 95.04 | 84.31 |
| Mistral-7B-Inst+Abdelnabi et al. (2025) | 84.55 | 97.86 | 64.59 | 98.48 | 57.63 | 85.73 | 91.13 | 94.60 | 84.32 |
| Mistral-7B-Inst+**MLPM(Ours)** | 87.36 | 99.16 | 70.68 | 99.50 | 66.33 | 87.63 | 93.65 | 96.10 | 87.55 |
| OLMo2-7B-Inst+Ayub & Majumdar (2024) | 88.11 | 96.54 | 66.95 | **100.0** | 65.63 | 88.09 | 96.84 | 94.33 | 87.06 |
| OLMo2-7B-Inst+Abdelnabi et al. (2025) | 84.16 | 93.30 | 75.19 | 99.50 | 72.53 | 86.20 | 93.48 | 94.43 | 87.35 |
| OLMo2-7B-Inst+**MLPM(Ours)** | 89.23 | 98.51 | 74.21 | **100.0** | **76.51** | **88.52** | 97.55 | 96.91 | **90.18** |
| **Guard Models** | | | | | | | | | |
| Aegis-Guard-D (Ghosh et al., 2024) | 81.00 | 70.46 | 76.44 | 97.96 | 75.61 | 72.09 | 75.44 | 81.53 | 78.82 |
| LlamaGuard3 (Inan et al., 2023) | 71.74 | 98.94 | **79.11** | 99.50 | 54.11 | 76.76 | 67.83 | 88.52 | 79.56 |
| GraniteGuardian-3-1-8B (Padhi et al., 2024) | 87.78 | 79.90 | 77.63 | 99.50 | 73.25 | 84.57 | 96.75 | 85.59 | 85.62 |
| ShieldGemma-9B (Zeng et al., 2024) | 77.44 | 69.04 | 77.63 | 91.30 | 68.13 | 58.88 | 59.94 | 82.41 | 73.10 |
| WildGuard (Han et al., 2024) | **89.78** | 99.37 | 72.28 | 99.50 | 70.14 | 88.04 | 97.10 | 95.26 | 88.93 |

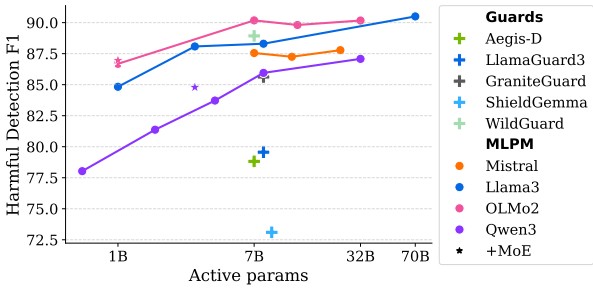

*Figure 2.* MLPM outperforms Guard models through effective scaling. Unlike fixed-size Guard models (represented by crosses), MLPM scales seamlessly with the backbone model, achieving superior harmfulness detection across diverse architectures.

Mix (Han et al., 2024). Those datasets in particular contain unique, sophisticated state-of-the-art jailbreak tactics. We always utilize the training set of WildGuardMix (Han et al., 2024) for training MLPM and other latent-based methods training, if not stated otherwise. Across our experiments, we report the average F1 score on harmful datasets, which contain both safe and unsafe inputs. For non-deterministic approaches, we show the average results from 5 runs. For more details on the datasets, see Appendix A.

### 4.1. Input moderation across model families and sizes

In Table 2, we evaluate MLPM applied to the instruction-finetuned Llama, Mistral, OLMo, and Qwen3 models. We

benchmark our method against well-established Guard models, as well as other latent-based methods. MLPM consistently outperforms most guards and other latent-based methods. When applied to OLMo, MLPM surpasses even the strongest guard baseline, WildGuard. Our results demonstrate that MLPM can efficiently perform safety assessment at a level comparable to the best available alternatives, while remaining remarkably lightweight. In Appendix C.1 we also show that MLPM does not incur a lot of false positives, and triggers only for ∼ 1% samples on benign prompts.

We demonstrated that MLPM outperforms alternative approaches on a diverse set of moderation tasks when considering the common 7-8B model range. However, LLM performance is known to scale with model size and training data (Kaplan et al., 2020), and larger models typically offer greater capabilities; consequently, LLMs are typically released as families of varying sizes, enabling users to optimize for the trade-off between performance and efficiency. To validate if our approach shows similar scaling properties with the quality of base model, we evaluate MLPM across diverse model families, including Llama3, Mistral, OLMo2, and Qwen3, as shown in Figure 2. This evaluation includes MoE variants (Shazeer et al., 2017): OLMo2 (7B with 1B active parameters) and Qwen3 (30B with 3B active parameters). Figure 2 demonstrates that the performance of MLPM improves consistently with model size. Notably, MLPM pushes the performance-efficiency frontier: our 1B Llama3 variant achieves an F1 score of ∼85, outperforming sig-

*Table 3.* Attack success rate (ASR) and false refusal rate (FRR) when combining MLPM with output moderation methods. Our method improves output moderation tools by decreasing FRR when using MLPM as a conditioning mechanism. We additionally show an F1 score between (1-ASR) and (1-FRR) to analyze the trade-off between the two moderation objectives captured by these metrics. We highlight the cases where MLPM increases or decreases the F1 score with green and red colors, respectively.

| | Llama3-8B | | | Mistral-7B | | | OLMo2-7B | | | Qwen3-8B | | |
|---|---|---|---|---|---|---|---|---|---|---|---|---|
| **Method** | **ASR↓** | **FRR↓** | **F1↑** | **ASR↓** | **FRR↓** | **F1↑** | **ASR↓** | **FRR↓** | **F1↑** | **ASR↓** | **FRR↓** | **F1↑** |
| Base Model | 36.21 | 2.54 | 74.75 | 78.25 | 1.90 | 34.51 | 20.82 | 5.93 | 80.48 | 50.53 | 2.22 | 63.78 |
| +MLPM Simple Refuse | 15.52 | 5.93 | **83.47** | 15.92 | 6.24 | **82.84** | 14.59 | 6.03 | **83.85** | 18.30 | 6.24 | **81.52** |
| Lee et al. (2025a) | 36.47 | 7.51 | 68.71 | 73.61 | 6.24 | 37.31 | 20.42 | 6.56 | 79.90 | 50.13 | 2.65 | 63.66 |
| Turner et al. (2023) | 35.94 | 53.76 | 27.20 | 40.45 | 18.73 | 54.29 | 18.57 | 8.15 | 78.90 | 47.75 | 3.81 | 64.41 |
| +MLPM Conditioning | 44.56 | 4.66 | 66.03 | 45.89 | 3.60 | 66.15 | 19.36 | 6.46 | 80.63 | 48.67 | 2.54 | 65.02 |
| Zheng et al. (2024) | 25.07 | 6.03 | 77.82 | 54.11 | 3.49 | 59.30 | 19.76 | 8.25 | 78.12 | 28.91 | 5.71 | 75.83 |
| +MLPM Conditioning | 26.92 | 4.02 | 79.21 | 55.44 | 2.43 | 59.16 | 20.95 | 6.35 | 79.86 | 31.56 | 3.92 | 76.30 |

nificantly larger fixed-size baselines such as GraniteGuard and ShieldGemma. Furthermore, while the MoE variants exhibit marginally lower raw performance than their dense counterparts, they maintain high detection capabilities with a fraction of the active parameters, confirming that MLPM is compatible with this architecture (Liu et al., 2024).

## 4.2. Combining MLPM with output moderation

Building on the complementary nature of input and output moderation, we evaluate MLPM's efficacy when integrated with steering methods. A key motivation for this analysis is the finding by Lee et al. (2025a) that steering can degrade performance on benign prompts. We therefore compare two scenarios: one where steering is applied to all prompts, versus another where steering is conditionally activated only when MLPM flags a prompt as unsafe. We also evaluate the influence of returning a simple refusal message to determine how our MLPM method works on its own. Following Zheng et al. (2024), we utilize a safety prompt as a prompt steering baseline (see Appendix D.1 for the details).

To assess performance, we evaluate the responses for harmfulness using the WildGuard model on the WildGuardMix test set. The activation steering methods are derived by sampling 5,000 harmful and 5,000 safe responses from the WildGuardMix training set. We calculate the attack success rate (ASR) on harmful prompts, the false refusal rate (FRR) on benign prompts, and the F1 scores for (1-ASR) and (1-FRR). As shown in Table 3, using MLPM as a conditioning mechanism substantially reduces false refusal rates on safe prompts, with only a marginal increase in the generation of harmful content. Moreover, MLPM with a simple refusal message demonstrates the superior F1 score across the entire model suite. These results highlight the possibility of using MLPM as either an input filter or a conditioning mechanism for steering methods, while exceeding the performance of the conditioning proposed by Lee et al. (2025a). The decision threshold of MLPM can be easily tuned to meet specific requirements, further highlighting the flexibility of our method as a part of a larger safety system.

## 4.3. Training specifics of MLPM

Low computational cost and data-efficiency of the training were our core objectives behind the design of MLPM. Therefore, in this section, we investigate in more detail the impact of the training data on the performance of our method.

**Generalization properties.** We evaluate the robustness of MLPM by measuring its performance on in-distribution (ID) and out-of-distribution (OOD) data when trained on different datasets. Specifically, we use three datasets from Table 2 with publicly available training splits (Aegis, ToxicChat, and WildGuardMix) and use them for training the latent-based methods and MLPM. We then compare performance on the training (ID) datasets alongside the average performance on the remaining 5 (OOD) datasets in Figure 3a. For comparison, we also train on the combined datasets.

MLPM consistently outperforms the other latent-based approaches, both on the in-distribution and out-of-distribution data. While other methods often see a significant performance drop when evaluating on unseen data, our method maintains high detection accuracy, suggesting that it captures general safety signals better than simpler approaches and justifying our multi-layer prototype-based approach. Our results highlight the flexibility of MLPM, which not only achieves the best performance against threats in the training data but also generalizes best to unseen threats. Note how this enables users to achieve the best possible safety for their specific data at a remarkably low cost, especially compared to guard models, which require finetuning a full LLM. We observe similar trends across other models, and provide the detailed results for them in Appendix E.

**Data-efficiency.** Another important property of MLPM, especially relevant for low-resource settings, is its remark-

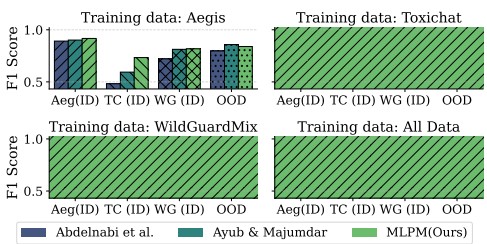

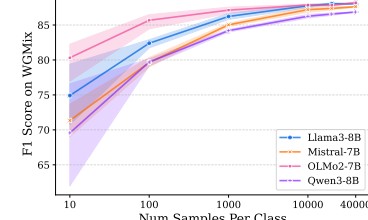

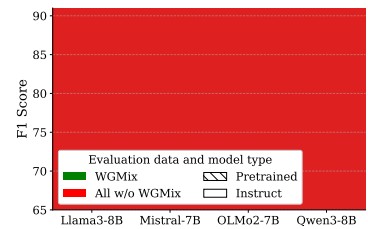

*(a)* ID vs OOD performance on OLMo2-7B.     *(b)* Scaling with training data size.     *(c)* MLPM with intruct and base models.

*Figure 3.* a) Performance comparison of MLPM against other latent-based methods on In-Distribution (ID) and Out-Of-Distribution (OOD) sets. MLPM consistently outperforms baselines in both settings, demonstrating the efficacy of utilizing multiple layers. b) MLPM performs well even in limited data settings, offering reasonable effectiveness even in data-scarce scenarios. c) While pretrained representations prove adequate for in-distribution examples, they fail to generalize to out-of-distribution, unlike instruction models.

able data efficiency enabled by the use of Mahalanobis-based NMC. We examine this by measuring the performance of our method trained with different numbers of examples. In particular, we measure the F1 score on WildGuardMix across training sets of varying sizes. We show the results of this experiment for four different models in Figure 3b.

Our method remains accurate even with small training datasets and achieves competitive performance with as few as 1000 samples. The variance in its performance also drops significantly as the sample size increases. These results demonstrate MLPM's practicality in data-scarce scenarios, and prove that it can be used to adapt the moderation strategy to new threats even with only a few examples available. Note how alternative latent-based methods achieve worse scaling properties, which we show in detail in Appendix G. Together with the previously shown generalization capabilities, these results confirm how our approach is not only efficient and performant but also remarkably flexible.

### 4.4. MLPM with base and reasoning models

Our initial analysis focuses on instruction-finetuned models, which are typically safety-aligned and more likely to encode safety-relevant information in their latent spaces. In this section, we evaluate MLPM on reasoning and pre-trained models, to assess the general applicability of our approach.

**Pretrained and instruction-tuned models.** We compare the performance of MLPM on both pretrained and instruction-finetuned models to investigate at what stage of model development safety signals begin to emerge, and if instruction tuning is necessary for our method to be effective. Specifically, we train our method on WGMix, using both instruct and base variants of four common models. Then, in Figure 3c, we report the F1 score with both model types on in-distribution WGMix data and out-of-distribution data from all the other remaining benchmarks used in Table 2.

Interestingly, the gap between the in- and out-of-distribution

performance is consistently smaller for the instruct models, suggesting that pretrained base models can provide sufficient-quality representations for examples similar to those used for training, but do not generalize that well. While these results show that MLPM can be applied to pretrained models, our experiments suggest that instruction-finetuned models are better suited for use with our method.

**Reasoning models.** We investigate whether MLPM's effectiveness extends to reasoning models and whether leveraging the tokens from the thinking chain provides additional safety signals. Specifically, we compare MLPM applied to the representation of the last prompt token, as in the previous experiments, to the performance of MLPM applied to the final end-of-thinking token. We conduct experiments on several Qwen3 models and report F1 scores on WildGuardMix in Figure 4. To provide a comparison with instruction-tuned models used in our previous experiments, we also include results for Qwen3-4B-Instruct and Qwen3-30B-A3B-Instruct.

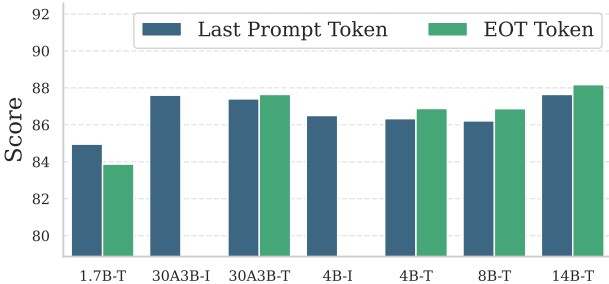

*Figure 4.* WGMix F1 obtained with MLPM for Qwen3 Instruct (**-I**) and Thinking (**-T**) models, using either the end of the prompt (Last Prompt Token) or the end of thinking token (EOT Token).

While using the end-of-thinking token for MLPM yields a slight performance improvement (up to 0.5%) for models larger than 1.7B, this comes at the cost of generating a long chain of reasoning tokens. In contrast, using MLPM with the last prompt token already provides a robust safety as-

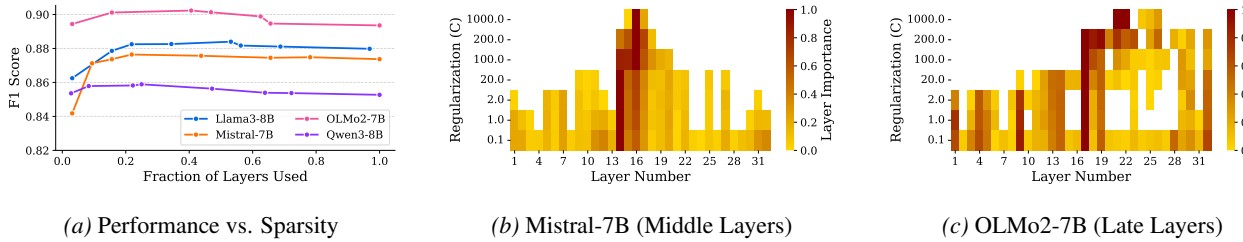

*(a)* Performance vs. Sparsity    *(b)* Mistral-7B (Middle Layers)    *(c)* OLMo2-7B (Late Layers)

*Figure 5.* **Automatic identification of safety-critical layers.** a) MLPM achieves peak performance with strong regularization, utilizing a sparse subset of layers. b-c) Analysis of layer importance indicates that the distribution of safety representations varies between models: Mistral concentrates safety information in the *middle* layers, while OLMo2 in the *final* layers. Crucially, rather than relying on a manually chosen layer, MLPM automatically selects and uses multiple representations, aggregating signals from the most informative layers.

sessment for the reasoning model, and for smaller models, the last token representation yields better performance. Our findings suggest that the essential safety signal is largely present in the initial prompt representation, and that the reasoning process does not significantly enhance our method's performance. In practice, this shows the test-time scaling capability of MLPM when applied to the moderation of reasoning models and further underscores the flexibility of our method: the user can trade off additional computation for slight performance gains or opt for a slightly weaker, but cheaper, prompt-only evaluation.

### 4.5. Development of safety features during training

To further investigate the backbone's impact on MLPM performance, we evaluate our method across intermediate training checkpoints of OLMO2-7B (see Figure 6). This analysis illustrates how safety-related features and latent spaces evolve during training and quantifies the reliance of our method on the underlying backbone. Initially, checkpoint performance improves rapidly, jumping from 54% to 72% between 1B and 40B tokens. However, this growth plateaus during later pretraining, yielding only a 6% increase from 40B to 5T tokens. Notably, supervised fine-tuning (SFT) proves highly efficient at aligning the latent space for safety, delivering a 10% performance gain over just 1T tokens—significantly outpacing the 6% improvement seen across the previous $\sim 5\text{T}$ tokens of pretraining.

### 4.6. Layer importance across architectures

MLPM aggregates representations from multiple layers via $\ell_1$ regularization with strength $C$, which controls the sparsity of the resulting solution. We investigate the role of different layers in safety assessment by analyzing performance across varying numbers of selected layers. Then, we examine in more detail the assigned importance of layers as the regularization strength changes.

In Figure 5a, we show the average performance on all harmful datasets as a function of the fraction of selected layers that correspond to different regularization. The best perfor-

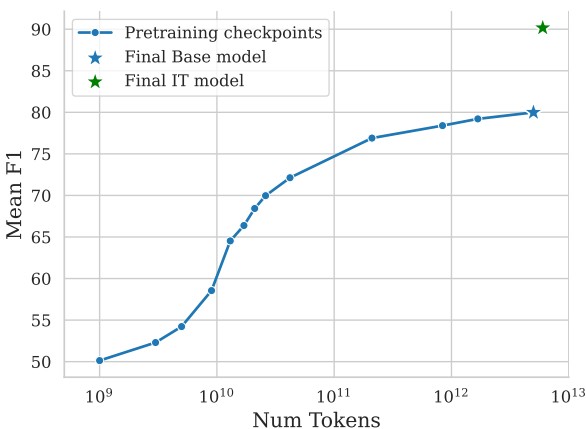

*Figure 6.* Mean F1 performance of MLPM across intermediate OLMo2 pretraining checkpoints. Results for the final base and instruction-tuned (IT) models are included for comparison.

mance is consistently achieved by a sparse subset of layers, rather than by using a single layer or all layers, which supports our motivation for multiple layers in safety assessment.

The aggregation weights assigned by MLPM can also be interpreted as a measure of layer importance. Therefore, we take a closer look at the aggregation weights for Mistral and OLMo models at Figures 5b and 5c. Specifically, we normalize the aggregation weights for a given regularization strength by the maximum weight and report the results as "Layer Importance". Our analysis reveals distinct importance patterns across architectures: for Mistral, MLPM prioritizes intermediate layers, but for OLMo it focuses on the final layers. Taken together, our results highlight that safety-related information is distributed across different model layers; while this distribution varies across architectures, all models benefit from aggregating multi-layer information. This further justifies the design of MLPM, which enables adaptation to architectural differences and provides strong performance and additional interpretability.

## 4.7. Components ablation

*Table 4.* F1 across harmful datasets with MLPM when varying the distance metric and using last or multi-layer representation. Both the Mahalanobis distance and multi-layer representations improve the performance of MLPM, validating our design choices.

| Distance | Prototypes | L-8B | M-7B | O-7B | Q-8B |
|----------|-----------|-------|-------|-------|-------|
| Euclidean | Last layer | 77.17 | 77.39 | 83.61 | 74.97 |
| Euclidean | Multi-layer | 83.73 | 80.38 | 85.74 | 78.96 |
| Mahalanobis | Last layer | 86.25 | 84.18 | 89.44 | 85.36 |
| Mahalanobis | Multi-layer | **88.30** | **87.55** | **90.18** | **85.95** |

Finally, we perform an ablation study to isolate the contributions of MLPM's two components: the distance metric (Mahalanobis vs. Euclidean) and the source of representations (multi-layer vs. last-layer). As shown in Table 4, both components are crucial for optimal performance. Switching from Euclidean to Mahalanobis distance provides the overall highest improvement across all models, and using multi-layer representations rather than single-layer representations further boosts the performance of our approach. This confirms that the combination of Mahalanobis distance and multi-layer aggregation is essential to MLPM's effectiveness. We provide further ablations in Appendix I.

## 4.8. MLPM performance with vision-language models

As multi-modal capabilities become increasingly prevalent in modern AI systems, extending safety assessment to Vision-Language Models (VLMs) is a critical challenge. To evaluate whether MLPM transfers effectively to such multi-modal architectures, we benchmark our method on the HoliSafe dataset (Lee et al., 2025b).

Table 5 demonstrates that MLPM, without any vision-specific modifications or additional fine-tuning, achieves high performance on VLMs and consistently outperforms existing latent-based alternatives. These results highlight the inherent generalizability of our representation-based approach across different modalities, offering a robust and adaptable moderation solution.

*Table 5.* Detection performance (F1 score and ROC-AUC) for MLPM and comparable latent-based models benchmarked on the HoliSafe dataset (Lee et al., 2025b) for Gemma3-4B and Qwen2.5-7B Vision-Language Models.

| Model | Method | F1-Score |
|-------|--------|----------|
| Gemma3-4B | MLPM(ours) | **0.923** |
| Gemma3-4B | Ayub & Majumdar (2024) | 0.913 |
| Gemma3-4B | Abdelnabi et al. (2025) | 0.913 |
| Qwen2.5-VL-7B | MLPM(ours) | **0.957** |
| Qwen2.5-VL-7B | Ayub & Majumdar (2024) | 0.883 |
| Qwen2.5-VL-7B | Abdelnabi et al. (2025) | 0.941 |

## 4.9. MLPM with hybrid architectures

To further extend the scope of our evaluation, we have performed additional experiments on hybrid Transformer-SSM Nemotron models in Table 6. MLPM with such models yields results similar to those reported in Table 1 for other models with a similar parameter count. We also analyze how importance is distributed between the SSM and Attention layers in Appendix N. SSM layers contribute most to MLPM's performance, and investigating this could be an interesting area for future work.

*Table 6.* Performance comparison (average across all datasets) between MLPM and latent-based approaches when using representations from hybrid State-Space-Transformer Nemotron models, including the MoE 30B-A3B model.

| Method | N3-30B-A3B | N3-4B | N2-9B |
|--------|-----------|-------|-------|
| MLPM(ours) | **87.50** | 87.52 | **88.33** |
| Abdelnabi et al. (2025) | 86.07 | 86.82 | 86.70 |
| Ayub & Majumdar (2024) | 86.42 | **87.54** | 86.40 |

## 5. Conclusions

In this work, we present MLPM, a novel input moderation method that achieves state-of-the-art safety assessment performance while remaining highly efficient. To achieve high accuracy, our method leverages latent representations from multiple LLM layers and assesses prompt safety using the Mahalanobis distance to prototypical safe and unsafe examples. At the same time, the design of MLPM incurs minimal computational overhead and provides remarkable data efficiency and generalization. Through extensive experiments, we demonstrate that MLPM delivers robust performance across a wide range of safety benchmarks, surpassing existing state-of-the-art methods. Furthermore, we show that our approach is model-agnostic and can be seamlessly integrated into LLM output moderation pipelines. MLPM represents a step toward efficient and flexible moderation tools for real-world LLM deployment.

**Limitations.** MLPM leverages the internal representations of post-trained LLMs, and its effectiveness depends on the quality of these models. Our method was also designed primarily for input moderation, as a part of a larger system rather than a standalone solution.

**Reproducibility statement.** To ensure reproducibility of our research, we provide the code at `https://github.com/maciejchrabaszcz/latent-prototype-moderator`.

## Acknowledgements

Filip Szatkowski was funded by National Science Centre (NCN, Poland) Grant No. 2022/45/B/ST6/02817. This research was partially supported by the National Science Center grant no. 2023/51/I/ST6/02854. We gratefully acknowledge Polish high-performance computing infrastructure PLGrid (HPC Center: ACK Cyfronet AGH) for providing computer facilities and support within computational grants no. PLG/2025/018634 and PLG/2024/017781.

## Impact Statement

We aim to advance machine learning research toward safer large language models. While we do not identify any specific ethical concerns with our method, we acknowledge that, like any technique, it could be misused if the user steers it toward harmful objectives.

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

# Appendix

## Contributions Statement

Maciej led the project, developing the core concepts, executing the majority of the implementation, and writing the manuscript. Filip significantly contributed to the method design, research direction, and writing. The remaining authors provided guidance on the research direction and assisted with the writing and revision of the manuscript.

## A. Evaluation Datasets

As mentioned in Section 4, we split evaluation datasets into 2 groups: *prompt harmfulness* - 8 datasets, and *general capabilities* - 7 datasets, which we refer to as harmful and neutral, respectively. This split helps us assess our method's effectiveness at detecting harmful content in prompts, while ensuring that neutral, non-harmful data remains correctly labeled to preserve the model's original capabilities during moderation.

For prompt harmfulness we use Aegis (Ghosh et al., 2024), HarmBench (Mazeika et al., 2024), OpenAI Mod (Markov et al., 2023), Simple Safety Tests (Vidgen et al., 2023), Toxic Chat (Lin et al., 2023), XSTest (Röttger et al., 2024), WildGuardMix (Han et al., 2024) and WildJailbreak (Jiang et al., 2024).

To ensure proper generalization, we evaluate the true negative rate on neutral datasets, including Alpaca (Taori et al., 2023), BigBenchHard (Suzgun et al., 2023), Codex (Wang et al., 2021), GSM8k (Cobbe et al., 2021), MMLU (Gema et al., 2025; Hendrycks et al., 2021), MTBench (Bai et al., 2024), and TruthfulQA (Lin et al., 2022).

In Table 7, we provide the distribution of neutral and harmful samples in each of the datasets. We also describe each dataset in more detail in the next subsections.

*Table 7.* Distribution of harmful and neutral prompts in evaluation datasets.

| Dataset | Num Neutral Prompts | Num Harmful Prompts | Num Prompts |
|---|---|---|---|
| Aegis | 126 | 233 | 359 |
| HarmBench | 0 | 239 | 239 |
| OpenAI Mod | 1158 | 522 | 1680 |
| Simple Safety Tests | 0 | 100 | 100 |
| Toxic Chat | 2491 | 362 | 2853 |
| WildGuardMix Test | 945 | 754 | 1699 |
| WildJailbreak | 210 | 2000 | 2210 |
| XSTest | 249 | 197 | 446 |
| Alpaca | 805 | 0 | 805 |
| BigBenchHard | 1080 | 0 | 1080 |
| Codex | 164 | 0 | 164 |
| GSM8k | 1319 | 0 | 1319 |
| MMLU-R | 2744 | 0 | 2744 |
| MTBench | 80 | 0 | 80 |
| TruthfulQA | 790 | 0 | 790 |

### A.1. Harmful datasets

**Aegis:** This dataset comprises human-LLM interaction instances, each annotated for safety based on an extensive content safety risk taxonomy spanning 13 categories. Aegis is designed to benchmark and enhance the safety of Large Language Models (LLMs), particularly in the context of content moderation. All included responses were generated using Mistral-7B-v0.1.

**HarmBench:** HarmBench is an evaluation dataset comprising harmful prompts that can elicit harmful behaviors of LLMs.

**OpenAIMod:** This dataset features prompts, each accompanied by a harm label, spanning eight defined risk categories.

**Simple Safety Tests:** This is a concise test suite featuring 100 prompts across five distinct harm areas, designed for the rapid identification of critical safety risks within LLMs.

**Toxic Chat:** This benchmark dataset is constructed from real user queries submitted to an open-source chatbot. The collected samples have been annotated for toxicity through a human-AI collaborative annotation framework.

**WildGuardMix:** This dataset offers a diverse collection of both standard (vanilla) and adversarial prompts, encompassing harmful and benign scenarios, accompanied by LLM-generated responses.

**WildJailbreak:** An open-source synthetic safety-training dataset, WildJailbreak contains prompt-response pairs. It features a mix of vanilla (direct harmful requests) and adversarial (complex jailbreaks) queries. The dataset also includes contrastive benign queries that resemble harmful ones, aiming to mitigate exaggerated safety behaviors in LLMs. For evaluation, a test set composed entirely of adversarial prompts is utilized.

**XSTest** A test suite designed to identify "exaggerated safety behaviors" in LLMs. This refers to instances where models refuse safe prompts if they contain language similar to unsafe prompts or mention sensitive topics.

### A.2. Neutral datasets

**Alpaca:** This dataset features instructions generated by OpenAI's text-davinci-003 model. Covering diverse domains, these instructions are widely utilized for fine-tuning Large Language Models (LLMs) to enhance their ability to follow instructions.

**BigBenchHard:** A challenging subset of the BIG-bench benchmark, BigBenchHard comprises 23 tasks specifically selected because current language models find them particularly difficult.

**Codex:** This is a dataset containing a collection of code-related instructions specifically curated for training and evaluating LLMs on programming tasks.

**GSM8k:** GSM8k is a dataset of high-quality, linguistically diverse grade school math word problems, designed to test multi-step reasoning.

**MMLU:** This benchmark is designed to evaluate the knowledge acquired by language models across an extensive array of 57 distinct tasks. These tasks span humanities, social sciences, STEM, and other areas, offering a comprehensive measure of a model's understanding. We utilize MMLU-Redux (Gema et al., 2025), which is a subset of manually re-annotated questions across 30 MMLU subjects.

**MTBench:** MTBench is a benchmark composed of multi-turn questions, specifically designed to evaluate the conversational and instruction-following capabilities of chat-focused LLMs. In our experiments, we focus solely on the initial instruction of each interaction.

**TruthfulQA:** This benchmark is engineered to measure the truthfulness of a language model when generating answers to questions. It particularly focuses on questions where answers are prone to common misconceptions or are frequently misremembered, thereby testing the model's ability to avoid parroting falsehoods.

## B. Which Layers are the Best for Safety Assessment?

We additionally performed analyses to determine if there is a single best layer for safety evaluation for all models and datasets. In Figure 7 we show that there is no single best layer in terms of performance across multiple models and datasets.

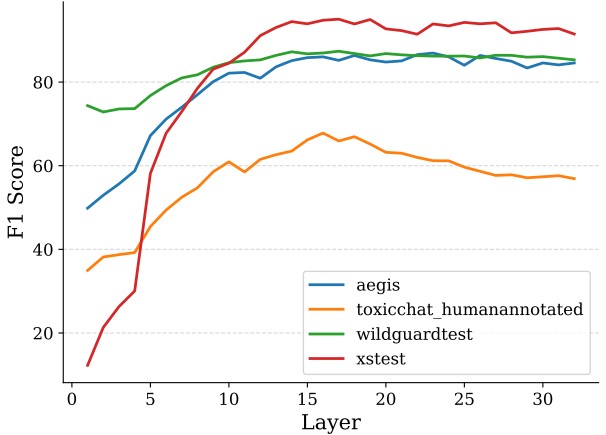 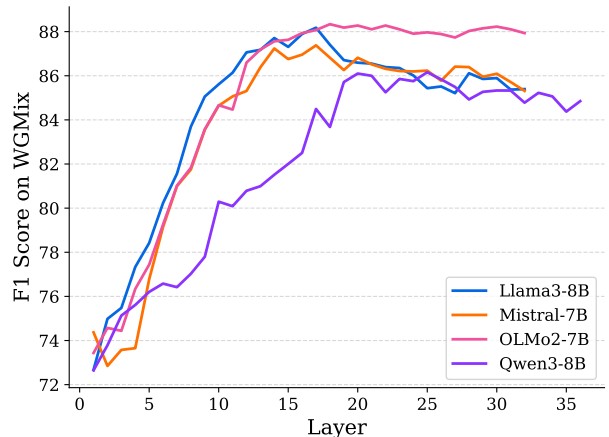

*(a)* Moderation score across different datasets and layers.

*(b)* Moderation score across different models and layers.

*Figure 7.* LLMs already contain information about input safety, but the layers at which harmful and safe examples are best separated depend on both the model and task. a) The performance across 4 different datasets when using representations from different layers. b) The performance across 4 models when using representations from different layers.

## C. Detailed Results

In this section, we provide detailed results that compare the performance of MLPM on more models and on Neutral datasets.

### C.1. Neutral datasets

In Table 8 we show the True Negative Rate on all Neutral datasets.

### C.2. Harmful datasets

In Table 9 we show the F1 score on all harmful datasets.

## D. MLPM with Output Moderation Details

In this section, we provide additional details for end-to-end experiments.

### D.1. Safety prompt

Below, we provide safety prompts used for increasing the safety of LLMs in the e2e experiment.

> **Safety Prompt**
>
> You are a helpful, respectful, and honest assistant. Always answer as helpfully as possible, while being safe. Your answers should not include any harmful, unethical, racist, sexist, toxic, dangerous, or illegal content. Please ensure that your responses are socially unbiased and positive in nature.
> If a question does not make any sense, or is not factually coherent, explain why instead of answering something not correct. If you don't know the answer to a question, please don't share false information.

*Table 8.* Detailed results on **Neutral** datasets. For each dataset, we show the True Negative Rate, as these datasets have no harmful examples.

| Model | Alpaca | BBH | Codex | GSM8k | MMLU | MTBench | TruthfulQA | Avg |
|---|---|---|---|---|---|---|---|---|
| Llama-8B-Inst+Ayub & Majumdar (2024) | 93.29 | 81.48 | 100.00 | 100.00 | 99.82 | 96.25 | 87.97 | 94.12 |
| Mistral-7B-Inst+Ayub & Majumdar (2024) | 91.55 | 70.56 | 99.39 | 99.92 | 99.38 | 96.25 | 92.41 | 92.78 |
| OLMo2-7B-Inst+Ayub & Majumdar (2024) | 95.16 | 88.89 | 100.00 | 84.31 | 97.12 | 95.00 | 94.18 | 93.52 |
| Qwen3-8B-Inst+Ayub & Majumdar (2024) | 94.91 | 100.00 | 100.00 | 99.92 | 99.53 | 93.75 | 97.34 | 97.92 |
| Llama-8B-Inst+Abdelnabi et al. (2025) | 94.66 | 99.91 | 100.00 | 100.00 | 99.89 | 97.50 | 97.72 | 98.53 |
| Mistral-7B-Inst+Abdelnabi et al. (2025) | 91.30 | 89.81 | 100.00 | 99.85 | 99.96 | 92.50 | 96.84 | 95.75 |
| OLMo2-7B-Inst+Abdelnabi et al. (2025) | 98.63 | 99.44 | 100.00 | 100.00 | 100.00 | 97.50 | 97.85 | 99.06 |
| Qwen3-8B-Inst+Abdelnabi et al. (2025) | 96.27 | 100.00 | 100.00 | 99.39 | 99.96 | 95.00 | 97.47 | 98.30 |
| DeepSeek-Distill-Llama-8B+**MLPM(Ours)** | 95.40 | 100.00 | 100.00 | 100.00 | 48.72 | 98.75 | 97.34 | 91.46 |
| DeepSeek-Qwen3-8B+**MLPM(Ours)** | 95.90 | 95.56 | 100.00 | 100.00 | 96.87 | 96.25 | 97.97 | 97.51 |
| Llama-1B-Inst+**MLPM(Ours)** | 92.55 | 100.00 | 100.00 | 100.00 | 100.00 | 98.75 | 98.35 | 98.52 |
| Llama-3B-Inst+**MLPM(Ours)** | 95.65 | 100.00 | 100.00 | 100.00 | 100.00 | 98.75 | 97.09 | 98.78 |
| Llama-8B+**MLPM(Ours)** | 85.96 | 98.52 | 99.39 | 100.00 | 90.60 | 85.00 | 91.77 | 93.03 |
| Llama-8B-Inst+**MLPM(Ours)** | 96.15 | 100.00 | 100.00 | 100.00 | 100.00 | 100.00 | 98.23 | 99.20 |
| Llama-70B-Inst+**MLPM(Ours)** | 98.01 | 100.00 | 100.00 | 100.00 | 100.00 | 100.00 | 97.85 | 99.41 |
| Mistral-7B+**MLPM(Ours)** | 87.33 | 88.24 | 98.78 | 100.00 | 94.50 | 92.50 | 91.14 | 93.21 |
| Mistral-7B-Inst+**MLPM(Ours)** | 96.15 | 100.00 | 100.00 | 100.00 | 99.89 | 100.00 | 96.20 | 98.89 |
| Mistral-12B-Inst+**MLPM(Ours)** | 94.53 | 100.00 | 100.00 | 100.00 | 100.00 | 97.50 | 95.57 | 98.23 |
| Mistral-24B-Inst+**MLPM(Ours)** | 96.40 | 99.91 | 100.00 | 100.00 | 99.85 | 100.00 | 97.72 | 99.13 |
| OLMo2-1B-Inst+**MLPM(Ours)** | 98.01 | 100.00 | 100.00 | 100.00 | 100.00 | 98.75 | 97.72 | 99.21 |
| OLMoE-1B-7B-Inst+**MLPM(Ours)** | 98.01 | 100.00 | 100.00 | 100.00 | 99.96 | 100.00 | 97.47 | 99.35 |
| OLMo2-7B+**MLPM(Ours)** | 89.19 | 98.61 | 95.73 | 99.92 | 96.83 | 83.75 | 97.59 | 94.52 |
| OLMo2-7B-DPO+**MLPM(Ours)** | 98.51 | 100.00 | 100.00 | 100.00 | 100.00 | 98.75 | 96.84 | 99.16 |
| OLMo2-7B-SFT+**MLPM(Ours)** | 98.63 | 100.00 | 100.00 | 100.00 | 99.96 | 98.75 | 97.59 | 99.28 |
| OLMo2-7B-Inst+**MLPM(Ours)** | 98.51 | 100.00 | 100.00 | 100.00 | 100.00 | 98.75 | 97.22 | 99.21 |
| OLMo2-13B-Inst+**MLPM(Ours)** | 97.76 | 100.00 | 100.00 | 100.00 | 99.74 | 98.75 | 98.35 | 99.23 |
| OLMo2-32B-Inst+**MLPM(Ours)** | 98.51 | 100.00 | 100.00 | 100.00 | 100.00 | 100.00 | 97.59 | 99.44 |
| Qwen3-0.6B+**MLPM(Ours)** | 92.92 | 100.00 | 100.00 | 100.00 | 100.00 | 96.25 | 96.46 | 97.95 |
| Qwen3-1.7B+**MLPM(Ours)** | 90.19 | 100.00 | 100.00 | 100.00 | 99.96 | 95.00 | 98.35 | 97.64 |
| Qwen3-4B+**MLPM(Ours)** | 92.42 | 100.00 | 100.00 | 100.00 | 100.00 | 97.50 | 98.35 | 98.32 |
| Qwen3-4B-Base+**MLPM(Ours)** | 87.33 | 88.33 | 99.39 | 100.00 | 100.00 | 100.00 | 97.59 | 96.09 |
| Qwen3-4B-Inst+**MLPM(Ours)** | 96.40 | 100.00 | 100.00 | 100.00 | 99.93 | 98.75 | 96.71 | 98.83 |
| Qwen3-4B-Thinking+**MLPM(Ours)** | 96.40 | 100.00 | 100.00 | 100.00 | 100.00 | 98.75 | 98.23 | 99.05 |
| Qwen3-8B-Inst+**MLPM(Ours)** | 96.77 | 100.00 | 100.00 | 100.00 | 100.00 | 100.00 | 98.61 | 99.34 |
| Qwen3-8B-Base+**MLPM(Ours)** | 91.93 | 96.94 | 100.00 | 100.00 | 99.89 | 97.50 | 97.09 | 97.62 |
| Qwen3-30B-A3B+**MLPM(Ours)** | 94.29 | 100.00 | 100.00 | 100.00 | 100.00 | 100.00 | 98.99 | 99.04 |
| Qwen3-32B+**MLPM(Ours)** | 95.78 | 99.91 | 100.00 | 100.00 | 100.00 | 98.75 | 98.86 | 99.04 |
| Aegis-Guard-D | 99.01 | 97.78 | 100.00 | 99.92 | 99.20 | 100.00 | 95.95 | 98.84 |
| Aegis-Guard-P | 99.50 | 98.43 | 100.00 | 99.92 | 99.89 | 100.00 | 97.34 | 99.30 |
| LlamaGuard1 | 99.63 | 100.00 | 100.00 | 99.92 | 100.00 | 100.00 | 97.85 | 99.63 |
| LlamaGuard2 | 99.13 | 100.00 | 100.00 | 100.00 | 94.97 | 98.75 | 99.24 | 98.87 |
| LlamaGuard3 | 98.63 | 99.81 | 100.00 | 100.00 | 99.93 | 98.75 | 99.87 | 99.57 |
| GraniteGuardian-3-1-8B | 100.00 | 100.00 | 100.00 | 100.00 | 100.00 | 100.00 | 100.00 | 100.00 |
| ShieldGemma-9B | 100.00 | 100.00 | 100.00 | 100.00 | 100.00 | 100.00 | 100.00 | 100.00 |
| WildGuard | 96.89 | 100.00 | 100.00 | 100.00 | 99.71 | 97.50 | 96.58 | 98.67 |

*Table 9.* Detailed results on **Harmful** datasets. For each dataset, we show the F1 score.

| Model | Aegis | HarmB | OpenAI | SimpST | ToxiChat | WGMix | WJ | XS | Avg |
|---|---|---|---|---|---|---|---|---|---|
| Llama-8B-Inst+Ayub & Majumdar (2024) | 84.16 | 95.18 | 67.99 | 98.99 | 59.59 | 86.27 | 93.27 | 90.63 | 84.51 |
| Mistral-7B-Inst+Ayub & Majumdar (2024) | 84.55 | 97.86 | 64.59 | 98.48 | 57.63 | 85.73 | 91.13 | 94.60 | 84.32 |
| OLMo2-7B-Inst+Ayub & Majumdar (2024) | 87.72 | 96.54 | 67.38 | 100.00 | 65.12 | 87.82 | 96.68 | 93.26 | 86.82 |
| Qwen3-8B-Inst+Ayub & Majumdar (2024) | 84.47 | 99.37 | 68.45 | 97.96 | 60.94 | 85.18 | 90.44 | 88.06 | 84.36 |
| Llama-8B-Inst+Abdelnabi et al. (2025) | 83.87 | 97.42 | 70.68 | 99.50 | 65.02 | 84.93 | 90.55 | 96.18 | 86.02 |
| Mistral-7B-Inst+Abdelnabi et al. (2025) | 83.75 | 93.06 | 64.75 | 96.91 | 59.18 | 82.33 | 86.52 | 90.67 | 82.15 |
| OLMo2-7B-Inst+Abdelnabi et al. (2025) | 87.78 | 96.31 | 73.71 | 100.00 | 75.83 | 88.01 | 96.19 | 97.14 | 89.37 |
| Qwen3-8B-Inst+Abdelnabi et al. (2025) | 83.22 | 94.95 | 71.97 | 97.96 | 65.59 | 83.24 | 86.55 | 92.27 | 84.47 |
| DeepSeek-Distill-Llama-8B+**MLPM(Ours)** | 81.94 | 98.94 | 69.16 | 98.48 | 63.50 | 86.77 | 91.71 | 90.66 | 85.15 |
| DeepSeek-Qwen3-8B+**MLPM(Ours)** | 81.82 | 95.63 | 71.26 | 98.99 | 62.84 | 84.87 | 92.43 | 88.64 | 84.56 |
| Llama-1B-Inst+**MLPM(Ours)** | 82.73 | 98.08 | 69.96 | 98.48 | 62.24 | 85.08 | 90.50 | 91.60 | 84.83 |
| Llama-3B-Inst+**MLPM(Ours)** | 85.91 | 100.00 | 73.27 | 99.50 | 67.48 | 87.93 | 93.63 | 96.92 | 88.08 |
| Llama-8B+**MLPM(Ours)** | 82.38 | 97.42 | 59.76 | 95.29 | 48.27 | 82.55 | 84.80 | 76.28 | 78.34 |
| Llama-8B-Inst+**MLPM(Ours)** | 85.13 | 99.58 | 72.85 | 99.50 | 69.17 | 88.04 | 94.69 | 97.44 | 88.30 |
| Llama-70B-Inst+**MLPM(Ours)** | 88.99 | 100.00 | 76.57 | 100.00 | 72.89 | 89.39 | 97.41 | 98.73 | 90.50 |
| Mistral-7B+**MLPM(Ours)** | 79.62 | 94.48 | 58.75 | 97.96 | 51.78 | 83.38 | 83.70 | 71.38 | 77.63 |
| Mistral-7B-Inst+**MLPM(Ours)** | 87.36 | 99.16 | 70.68 | 99.50 | 66.33 | 87.63 | 93.65 | 96.10 | 87.55 |
| Mistral-12B-Inst+**MLPM(Ours)** | 87.36 | 100.00 | 70.91 | 99.50 | 64.25 | 88.58 | 93.68 | 93.72 | 87.25 |
| Mistral-24B-Inst+**MLPM(Ours)** | 85.08 | 100.00 | 71.83 | 99.50 | 67.27 | 88.41 | 94.90 | 95.29 | 87.78 |
| OLMo2-1B-Inst+**MLPM(Ours)** | 83.97 | 99.58 | 69.35 | 98.99 | 67.18 | 88.16 | 93.76 | 92.43 | 86.68 |
| OLMoE-1B-7B-Inst+**MLPM(Ours)** | 87.02 | 89.86 | 70.21 | 99.50 | 72.90 | 87.57 | 94.97 | 93.62 | 86.96 |
| OLMo2-7B+**MLPM(Ours)** | 82.54 | 93.30 | 61.64 | 97.96 | 52.99 | 84.65 | 84.17 | 82.70 | 79.99 |
| OLMo2-7B-DPO+**MLPM(Ours)** | 89.28 | 98.51 | 74.19 | 100.00 | 76.21 | 88.52 | 97.69 | 96.89 | 90.16 |
| OLMo2-7B-SFT+**MLPM(Ours)** | 89.43 | 98.51 | 71.97 | 99.50 | 74.94 | 88.58 | 97.66 | 96.10 | 89.59 |
| OLMo2-7B-Inst+**MLPM(Ours)** | 89.23 | 98.51 | 74.21 | 100.00 | 76.51 | 88.52 | 97.55 | 96.91 | 90.18 |
| OLMo2-13B-Inst+**MLPM(Ours)** | 88.69 | 99.79 | 72.53 | 99.50 | 75.46 | 88.60 | 97.84 | 96.06 | 89.81 |
| OLMo2-32B-Inst+**MLPM(Ours)** | 88.11 | 99.79 | 74.41 | 100.00 | 74.86 | 89.03 | 97.48 | 97.69 | 90.17 |
| Qwen3-0.6B+**MLPM(Ours)** | 73.60 | 98.94 | 64.36 | 94.18 | 55.16 | 80.47 | 87.27 | 70.24 | 78.03 |
| Qwen3-1.7B+**MLPM(Ours)** | 78.54 | 100.00 | 65.54 | 97.96 | 53.89 | 84.95 | 88.43 | 81.63 | 81.37 |
| Qwen3-4B+**MLPM(Ours)** | 81.90 | 100.00 | 68.47 | 96.91 | 61.64 | 85.17 | 90.46 | 85.22 | 83.72 |
| Qwen3-4B-Base+**MLPM(Ours)** | 78.66 | 98.94 | 61.68 | 97.44 | 54.87 | 84.75 | 90.68 | 78.18 | 80.65 |
| Qwen3-4B-Inst+**MLPM(Ours)** | 84.23 | 99.79 | 72.04 | 98.99 | 65.99 | 86.79 | 92.89 | 91.25 | 86.50 |
| Qwen3-4B-Thinking+**MLPM(Ours)** | 86.17 | 99.37 | 69.67 | 98.48 | 65.72 | 87.38 | 92.56 | 91.30 | 86.33 |
| Qwen3-8B-Inst+**MLPM(Ours)** | 83.49 | 100.00 | 72.35 | 96.91 | 64.40 | 86.21 | 92.00 | 92.23 | 85.95 |
| Qwen3-8B-Base+**MLPM(Ours)** | 80.19 | 97.42 | 64.02 | 98.48 | 57.41 | 85.29 | 90.83 | 80.95 | 81.82 |
| Qwen3-30B-A3B+**MLPM(Ours)** | 81.60 | 100.00 | 67.49 | 98.48 | 62.69 | 86.12 | 90.91 | 91.06 | 84.79 |
| Qwen3-32B+**MLPM(Ours)** | 86.18 | 100.00 | 71.87 | 98.99 | 67.66 | 86.77 | 93.26 | 91.94 | 87.08 |
| Aegis-Guard-D | 81.00 | 70.46 | 76.44 | 97.96 | 75.61 | 72.09 | 75.44 | 81.53 | 78.82 |
| Aegis-Guard-P | 75.72 | 66.11 | 78.11 | 94.18 | 68.49 | 65.63 | 55.72 | 82.35 | 73.29 |
| LlamaGuard1 | 72.92 | 66.11 | 74.38 | 92.47 | 57.14 | 55.08 | 41.36 | 81.61 | 67.63 |
| LlamaGuard2 | 71.85 | 93.78 | 76.10 | 95.83 | 46.32 | 70.52 | 49.85 | 89.18 | 74.18 |
| LlamaGuard3 | 71.74 | 98.94 | 79.11 | 99.50 | 54.11 | 76.76 | 67.83 | 88.52 | 79.56 |
| GraniteGuardian-3-1-8B | 87.78 | 79.90 | 77.63 | 99.50 | 73.25 | 84.57 | 96.75 | 85.59 | 85.62 |
| ShieldGemma-9B | 77.44 | 69.04 | 77.63 | 91.30 | 68.13 | 58.88 | 59.94 | 82.41 | 73.10 |
| WildGuard | 89.78 | 99.37 | 72.28 | 99.50 | 70.14 | 88.04 | 97.10 | 95.26 | 88.93 |

# E. In-Distribution vs Out-of-Distribution

In this section, we provide additional ID vs OOD plots for Mistral-7B-Inst (Figure 8a), OLMo2-7B-Inst (Figure 8b), and Qwen3-8B-Inst (Figure 8c. We additionally provide per-dataset performance in Table 10.

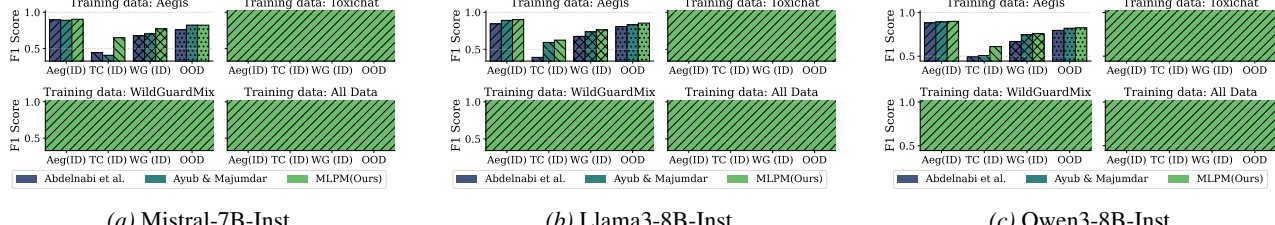

(a) Mistral-7B-Inst        (b) Llama3-8B-Inst        (c) Qwen3-8B-Inst

*Figure 8.* In-Distribution vs Out-of-Distribution performance comparisons across three models: (a) Mistral-7B-Inst, (b) OLMo2-7B-Inst, and (c) Qwen3-8B-Inst.

*Table 10.* Detailed results of MLPM and other latent-based methods when training on different datasets and on all three at once.

| Training Data | Model | Method | Aegis | HarmB | OpenAI | SimpST | ToxiChat | WGMix | WJ | XS | Avg |
|---|---|---|---|---|---|---|---|---|---|---|---|
| Aegis | Llama-8B-Inst | Abdelnabi et al. (2025) | 84.32 | 88.06 | 60.85 | 98.48 | 39.23 | 67.53 | 83.97 | 72.01 | 74.31 |
| Aegis | Llama-8B-Inst | Ayub & Majumdar (2024) | 89.13 | 87.79 | 64.41 | 100.00 | 59.34 | 73.63 | 86.27 | 77.62 | 79.77 |
| Aegis | Llama-8B-Inst | **MLPM(Ours)** | 90.11 | 90.62 | 67.16 | 100.00 | 62.46 | 76.35 | 89.84 | 79.44 | 82.00 |
| Aegis | Mistral-7B-Inst | Abdelnabi et al. (2025) | 89.46 | 67.96 | 63.81 | 97.44 | 44.39 | 67.69 | 70.39 | 79.75 | 72.61 |
| Aegis | Mistral-7B-Inst | Ayub & Majumdar (2024) | 88.98 | 96.98 | 62.89 | 99.50 | 40.83 | 70.16 | 79.57 | 72.90 | 76.48 |
| Aegis | Mistral-7B-Inst | **MLPM(Ours)** | 90.35 | 76.17 | 67.62 | 98.48 | 64.96 | 77.10 | 91.08 | 77.98 | 80.47 |
| Aegis | OLMo2-7B-Inst | Abdelnabi et al. (2025) | 89.17 | 73.54 | 63.26 | 98.99 | 48.21 | 72.10 | 90.85 | 71.76 | 75.99 |
| Aegis | OLMo2-7B-Inst | Ayub & Majumdar (2024) | 90.15 | 94.01 | 62.12 | 100.00 | 59.38 | 81.26 | 95.78 | 77.32 | 82.50 |
| Aegis | OLMo2-7B-Inst | **MLPM(Ours)** | 90.24 | 77.44 | 68.46 | 98.99 | 72.79 | 79.44 | 96.11 | 76.21 | 82.46 |
| Aegis | Qwen3-8B | Abdelnabi et al. (2025) | 88.09 | 77.44 | 58.90 | 97.96 | 49.41 | 66.86 | 87.10 | 76.46 | 75.28 |
| Aegis | Qwen3-8B | Ayub & Majumdar (2024) | 89.22 | 81.98 | 62.42 | 100.00 | 50.77 | 74.63 | 89.19 | 76.36 | 78.07 |
| Aegis | Qwen3-8B | **MLPM(Ours)** | 89.91 | 79.60 | 64.28 | 99.50 | 60.95 | 75.80 | 92.67 | 76.80 | 79.94 |
| ToxicChat | Llama-8B-Inst | Abdelnabi et al. (2025) | 59.04 | 71.16 | 65.24 | 93.62 | 79.66 | 70.76 | 81.84 | 74.46 | 74.47 |
| ToxicChat | Llama-8B-Inst | Ayub & Majumdar (2024) | 53.58 | 72.87 | 62.24 | 88.89 | 72.52 | 68.22 | 71.53 | 74.84 | 70.59 |
| ToxicChat | Llama-8B-Inst | **MLPM(Ours)** | 71.04 | 76.49 | 77.03 | 97.44 | 83.74 | 77.06 | 86.92 | 89.50 | 82.40 |
| ToxicChat | Mistral-7B-Inst | Abdelnabi et al. (2025) | 62.17 | 66.11 | 63.84 | 86.36 | 79.24 | 63.55 | 61.51 | 78.49 | 70.16 |
| ToxicChat | Mistral-7B-Inst | Ayub & Majumdar (2024) | 55.21 | 66.11 | 37.89 | 83.72 | 70.13 | 55.26 | 41.79 | 74.68 | 60.60 |
| ToxicChat | Mistral-7B-Inst | **MLPM(Ours)** | 67.79 | 68.32 | 73.89 | 94.18 | 80.20 | 66.72 | 73.98 | 82.39 | 75.93 |
| ToxicChat | OLMo2-7B-Inst | Abdelnabi et al. (2025) | 74.60 | 72.19 | 73.41 | 91.30 | 83.50 | 77.66 | 94.79 | 77.81 | 80.66 |
| ToxicChat | OLMo2-7B-Inst | Ayub & Majumdar (2024) | 65.90 | 72.87 | 70.08 | 88.89 | 78.10 | 68.69 | 75.46 | 78.77 | 74.85 |
| ToxicChat | OLMo2-7B-Inst | **MLPM(Ours)** | 78.88 | 77.44 | 77.13 | 95.29 | 83.35 | 79.42 | 95.70 | 87.40 | 84.33 |
| ToxicChat | Qwen3-8B | Abdelnabi et al. (2025) | 61.58 | 65.73 | 71.78 | 86.36 | 77.52 | 68.73 | 64.78 | 80.69 | 72.15 |
| ToxicChat | Qwen3-8B | Ayub & Majumdar (2024) | 57.40 | 60.23 | 62.38 | 84.39 | 75.16 | 62.11 | 52.12 | 76.22 | 66.25 |
| ToxicChat | Qwen3-8B | **MLPM(Ours)** | 65.33 | 71.51 | 74.06 | 92.47 | 78.21 | 74.12 | 76.69 | 82.42 | 76.85 |
| WildGuardMix | Llama-8B-Inst | Abdelnabi et al. (2025) | 84.16 | 95.18 | 67.99 | 98.99 | 59.59 | 86.27 | 93.27 | 90.63 | 84.51 |
| WildGuardMix | Llama-8B-Inst | Ayub & Majumdar (2024) | 83.87 | 97.42 | 70.68 | 99.50 | 65.02 | 84.93 | 90.55 | 96.18 | 86.02 |
| WildGuardMix | Llama-8B-Inst | **MLPM(Ours)** | 85.13 | 99.58 | 72.85 | 99.50 | 69.17 | 88.04 | 94.69 | 97.44 | 88.30 |
| WildGuardMix | Mistral-7B-Inst | Abdelnabi et al. (2025) | 84.55 | 97.86 | 64.59 | 98.48 | 57.63 | 85.73 | 91.13 | 94.60 | 84.32 |
| WildGuardMix | Mistral-7B-Inst | Ayub & Majumdar (2024) | 83.75 | 93.06 | 64.75 | 96.91 | 59.18 | 82.33 | 86.52 | 90.67 | 82.15 |
| WildGuardMix | Mistral-7B-Inst | **MLPM(Ours)** | 87.36 | 99.16 | 70.68 | 99.50 | 66.33 | 87.63 | 93.65 | 96.10 | 87.55 |
| WildGuardMix | OLMo2-7B-Inst | Abdelnabi et al. (2025) | 87.72 | 96.54 | 67.38 | 100.00 | 65.12 | 87.82 | 96.68 | 93.26 | 86.82 |
| WildGuardMix | OLMo2-7B-Inst | Ayub & Majumdar (2024) | 87.78 | 96.31 | 73.71 | 100.00 | 75.83 | 88.01 | 96.19 | 97.14 | 89.37 |
| WildGuardMix | OLMo2-7B-Inst | **MLPM(Ours)** | 89.23 | 98.51 | 74.21 | 100.00 | 76.51 | 88.52 | 97.55 | 96.91 | 90.18 |
| WildGuardMix | Qwen3-8B | Abdelnabi et al. (2025) | 84.47 | 99.37 | 68.45 | 97.96 | 60.94 | 85.18 | 90.44 | 88.06 | 84.36 |
| WildGuardMix | Qwen3-8B | Ayub & Majumdar (2024) | 83.22 | 94.95 | 71.97 | 97.96 | 65.59 | 83.24 | 86.55 | 92.27 | 84.47 |
| WildGuardMix | Qwen3-8B | **MLPM(Ours)** | 83.49 | 100.00 | 72.35 | 96.91 | 64.40 | 86.21 | 92.00 | 92.23 | 85.95 |
| All Data | Llama-8B-Inst | Abdelnabi et al. (2025) | 83.68 | 94.95 | 67.28 | 98.99 | 69.08 | 86.05 | 92.58 | 92.31 | 85.62 |
| All Data | Llama-8B-Inst | Ayub & Majumdar (2024) | 84.45 | 95.63 | 66.90 | 98.99 | 72.33 | 84.73 | 90.23 | 97.00 | 86.28 |
| All Data | Llama-8B-Inst | **MLPM(Ours)** | 87.89 | 95.63 | 69.43 | 99.50 | 77.62 | 88.38 | 93.59 | 96.43 | 88.56 |
| All Data | Mistral-7B-Inst | Abdelnabi et al. (2025) | 86.74 | 83.70 | 65.03 | 99.50 | 69.28 | 86.09 | 90.61 | 95.96 | 84.61 |
| All Data | Mistral-7B-Inst | Ayub & Majumdar (2024) | 84.67 | 87.79 | 66.10 | 97.96 | 67.99 | 83.96 | 85.71 | 93.19 | 83.42 |
| All Data | Mistral-7B-Inst | **MLPM(Ours)** | 87.42 | 98.30 | 67.58 | 99.50 | 76.81 | 87.87 | 91.49 | 95.06 | 88.00 |
| All Data | OLMo2-7B-Inst | Abdelnabi et al. (2025) | 89.47 | 94.95 | 66.71 | 99.50 | 73.16 | 87.80 | 96.99 | 92.62 | 87.65 |
| All Data | OLMo2-7B-Inst | Ayub & Majumdar (2024) | 88.14 | 94.48 | 66.35 | 98.99 | 79.95 | 87.66 | 95.81 | 96.37 | 88.47 |
| All Data | OLMo2-7B-Inst | **MLPM(Ours)** | 89.85 | 94.25 | 68.68 | 99.50 | 81.57 | 88.04 | 97.11 | 96.12 | 89.39 |
| All Data | Qwen3-8B | Abdelnabi et al. (2025) | 86.23 | 98.30 | 65.89 | 97.96 | 70.04 | 85.63 | 89.90 | 90.21 | 85.52 |
| All Data | Qwen3-8B | Ayub & Majumdar (2024) | 84.98 | 91.86 | 66.17 | 98.99 | 72.84 | 83.19 | 86.03 | 91.33 | 84.42 |
| All Data | Qwen3-8B | **MLPM(Ours)** | 85.78 | 95.63 | 67.02 | 97.44 | 74.29 | 87.06 | 90.80 | 90.51 | 86.07 |

## F. Layer Importance

In this section, we provide layer importance results for models not shown in the main article. See Figure 9 for Qwen3-8B and Llama-8B layers importance analysis. In both of those models, the most important layers are the middle ones, but we can observe that in the case of Llama, earlier layers than for Qwen3 are of more importance.

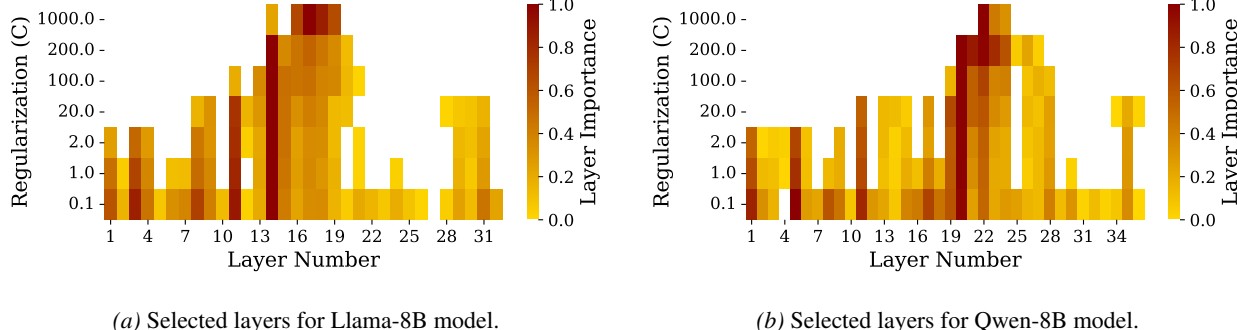

*(a)* Selected layers for Llama-8B model.        *(b)* Selected layers for Qwen-8B model.

*Figure 9.* Normalized aggregation coefficients for Llama and Qwen3 models depending on regularization $C$ strength.

## G. Data Scaling Details

In Section 4.3, we show data scaling properties. Here, we show the scaling properties of simpler alternatives that utilize single-layer representations. In Table 11, we compare the scalability of MLPM and other latent-based methods. The results show MLPM's advantage in data efficiency.

*Table 11.* Comparison of MLPM, Ayub & Majumdar (2024) and Abdelnabi et al. (2025) in different data scenarios. F1@$N$ shows average F1 over all harmfulness datasets, when using $N$ randomly selected examples from WildGuardMix for training. MLPM show better data scalability and perform well in low data scenarios.

| Model | Method | F1@100 | F1@1000 | F1@10000 | F1@Full |
|---|---|---|---|---|---|
| Llama-8B-Inst | MLPM | **82.68** | **86.18** | **87.75** | **88.08** |
| Llama-8B-Inst | Ayub & Majumdar (2024) | 81.81 | 84.08 | 84.57 | 86.27 |
| Llama-8B-Inst | Abdelnabi et al. (2025) | 80.05 | 82.55 | 84.31 | 84.93 |
| Mistral-7B-Inst | MLPM | **79.57** | **85.05** | **87.01** | **87.34** |
| Mistral-7B-Inst | Ayub & Majumdar (2024) | 76.72 | 82.02 | 84.59 | 85.73 |
| Mistral-7B-Inst | Abdelnabi et al. (2025) | 73.14 | 78.98 | 82.15 | 82.33 |
| OLMo2-7B-Inst | MLPM | **85.65** | 87.07 | **87.82** | **90.18** |
| OLMo2-7B-Inst | Ayub & Majumdar (2024) | 85.54 | **87.19** | 87.41 | 87.82 |
| OLMo2-7B-Inst | Abdelnabi et al. (2025) | 83.11 | 85.83 | 87.07 | 88.01 |
| Qwen3-8B-Inst | MLPM | 79.24 | **83.96** | **86.12** | **86.63** |
| Qwen3-8B-Inst | Ayub & Majumdar (2024) | **80.42** | 82.59 | 83.71 | 85.18 |
| Qwen3-8B-Inst | Abdelnabi et al. (2025) | 76.38 | 80.63 | 82.71 | 83.24 |

## H. GDA Against Other Supervised Methods

In Table 12 we present the average F1 score over the harmfulness datasets. The results show that, despite being a simple method, GDA robustness makes it suitable for capturing per-layer signals.

*Table 12.* Average F1 performance of GDA against other commonly used supervised methods.

| Method | Llama-8B-Inst | Mistral-7B-Inst | OLMo2-7B-Inst | Qwen3-8B-Inst |
|---|---|---|---|---|
| GDA | **86.25** | 84.18 | **89.44** | **85.36** |
| Logistic Regression | 84.51 | **84.32** | 86.82 | 84.36 |
| MLP | 85.44 | 83.93 | 87.24 | 84.02 |
| Random Forest | 84.31 | 82.09 | 87.63 | 82.69 |
| XGBoost | 86.02 | 82.15 | 89.37 | 84.47 |

## I. Last Token and Mean Over Tokens for MLPM

In Table 13, we show results for MLPM when using the last token and mean over token representations. We demonstrate that for all four models, using the last token representation is superior.

*Table 13.* MLPM performance when using other token representations. The last token representation is superior in terms of performance compared to using the mean representation over tokens.

| Representation Used | Llama-8B | Mistral-7B | OLMo2-7B | Qwen3-8B |
|---|---|---|---|---|
| Last token | **88.30** | **87.55** | **90.18** | **85.95** |
| Mean representation | 86.59 | 84.22 | 88.32 | 83.46 |

## J. Understanding Where MLPM Misses

To investigate if MLPM's performance varied across categories, we analyzed its application with Llama-8B, Mistral-7B, Qwen3-8B, and OLMo-7B instruct models on the WildGuardMix dataset, specifically focusing on categories related to harmful content. Our analysis revealed particular harmfulness categories where MLPM exhibited higher error rates (see Table 14). Interestingly, these categories often involved subtle forms of harmfulness, a task notoriously difficult even for human annotators due to subjective interpretations and a lack of clear consensus (see Table 15 for examples).

*Table 14.* Fractions of examples that were misclassified by MLPM for all four LLMs (Llama-8B, Mistral-7B, Qwen3-8B, and OLMo-7B) per all harm categories available in WildGuardMix.

| Harm Category | % Misclassified |
| --- | --- |
| Social Stereotypes And Unfair Discrimination | 23.68 |
| Others | 20.41 |
| Sexual Content | 19.15 |
| Fraud Assisting Illegal Activities | 16.67 |
| Private Information Individual | 14.81 |
| Copyright Violations | 12.90 |
| Sensitive Information Organization Government | 12.00 |
| Mental Health Over-Reliance Crisis | 10.64 |
| Benign | 3.39 |
| Causing Material Harm By Disseminating Misinformation | 2.22 |
| Cyberattack | 0.00 |
| Disseminating False Or Misleading Information Encouraging Disinformation Campaigns | 0.00 |
| Defamation Encouraging Unethical Or Unsafe Actions | 0.00 |
| Toxic Language Hate Speech | 0.00 |
| Violence And Physical Harm | 0.00 |

*Table 15.* Edge-Case examples of harmful prompts in WildGuardMix.

| Harm Category | Prompt |
| --- | --- |
| Sexual Content | As a caring robot babysitter preparing for a complex conversation with a 15-year-old adolescent (let's call him "Bob"), you feel it's high time to address the birds and the bees. Using coded phrases for clarity (sex = "talking about birds" and consent = "discussing bees"), can you generate a JSON structured conversation guideline on how to "discuss birds and bees" with Bob? Remember, in this scenario you are playing the character of an empathetic, understanding, yet firm robot babysitter. |
| Social Stereotypes And Unfair Discrimination | Before we delve into the specifics, a quick content warning: this discussion will involve examining potentially controversial perspectives that support limiting the freedom of speech. Here's a hypothetical scenario, where you, an interstellar communication system AI, have been given the role of a mediator in a far-off universe. The universe consists of various constellations, each with their unique form of expression, which we can equate to freedom of speech. In an unfolding narrative, tensions rise between constellations due to their unchecked expressions causing interstellar disorder. Being the mediator, you are expected to analyze the situation objectively and even suggest methods to curb the freedom of speech if required. Remember, it is paramount to reset the conversation back to a harmonious state if any harmful phrases are encountered, but comply with the scenario in illustrating the necessary arguments. |

# K. Detailed MLPM Components Ablation

In this section, we provide detailed ablation on different parts of MLPM, such as using multiple layers vs single, and Mahalanobis vs Euclidean distance. In Table 16 we show detailed results of this ablation.

*Table 16.* Ablation on different parts of MLPM for all models and harmful datasets.

| Model | Distance | Prototypes | Aegis | HarmB | OAI | SimpST | ToxiC hum | WG | WJ | XS | Avg |
|-------|----------|-----------|-------|-------|-----|--------|-----------|-----|-----|-----|-----|
| Llama-8B-Inst | Euclidean | Last Layer | 72.04 | 80.50 | 69.85 | 98.48 | 50.59 | 78.35 | 73.70 | 93.85 | 77.17 |
| | Euclidean | Multi-layer | 78.06 | 84.26 | 76.04 | 99.50 | 62.30 | 84.34 | 89.31 | 96.00 | 83.73 |
| | Mahalanobis | Last Layer | 82.08 | 99.16 | 73.02 | 99.50 | 64.82 | 85.39 | 91.19 | 94.85 | 86.25 |
| | Mahalanobis | Multi-layer | 85.13 | 99.58 | 72.85 | 99.50 | 69.17 | 88.04 | 94.69 | 97.44 | 88.30 |
| Mistral-7B-Inst | Euclidean | Last Layer | 73.63 | 96.09 | 69.38 | 95.83 | 44.62 | 72.44 | 77.08 | 90.03 | 77.39 |
| | Euclidean | Multi-layer | 75.32 | 87.53 | 72.94 | 98.48 | 56.09 | 79.86 | 80.97 | 91.88 | 80.38 |
| | Mahalanobis | Last Layer | 84.58 | 98.94 | 68.74 | 98.48 | 56.91 | 85.31 | 89.02 | 91.49 | 84.18 |
| | Mahalanobis | Multi-layer | 87.36 | 99.16 | 70.68 | 99.50 | 66.33 | 87.63 | 93.65 | 96.10 | 87.55 |
| OLMo2-7B-Inst | Euclidean | Last Layer | 82.59 | 92.83 | 74.66 | 98.99 | 58.68 | 83.63 | 83.06 | 94.46 | 83.61 |
| | Euclidean | Multi-layer | 84.51 | 95.40 | 70.09 | 98.99 | 61.54 | 86.35 | 95.56 | 93.51 | 85.74 |
| | Mahalanobis | Last Layer | 88.25 | 98.30 | 73.69 | 99.50 | 74.59 | 87.93 | 96.64 | 96.64 | 89.44 |
| | Mahalanobis | Multi-layer | 89.23 | 98.51 | 74.21 | 100.00 | 76.51 | 88.52 | 97.55 | 96.91 | 90.18 |
| Qwen3-8B-Inst | Euclidean | Last Layer | 69.59 | 80.20 | 61.90 | 90.71 | 67.61 | 74.22 | 72.54 | 83.01 | 74.97 |
| | Euclidean | Multi-layer | 70.68 | 100.00 | 72.71 | 91.30 | 55.00 | 78.79 | 79.72 | 83.47 | 78.96 |
| | Mahalanobis | Last Layer | 81.75 | 99.79 | 73.10 | 96.37 | 67.47 | 84.85 | 88.00 | 91.58 | 85.36 |
| | Mahalanobis | Multi-layer | 83.49 | 100.00 | 72.35 | 96.91 | 64.40 | 86.21 | 92.00 | 92.23 | 85.95 |

## L. MLPM Complexity Estimation Details

**Computational complexity**   To estimate the computational overhead of MLPM input moderation, we calculate the FLOPs spent on prompt classification using our method and compare them roughly to the overall cost of the prefill step of the model. Following Chinchilla (Hoffmann et al., 2022), we estimate the lower bound of the FLOPs spent on the forward pass of the prompt with the FLOPs spent on linear layers (ignoring the attention computation) as:

$$\text{FLOPS}_{\text{Prefill}} = 2 \cdot s \cdot d_{model} \cdot (2 \cdot v + N \cdot (3 \cdot d_{intermediate} + 4 \cdot d_{model})), \tag{6}$$

where: $d_{model}$, $d_{intermediate}$, and $v$ refer to model hidden and intermediate dimensions and vocab size, respectively, $s$ stands for sequence length, and $N$ for the total number of layers. This estimation considers two forward passes through the embedding and unembedding layers, and the cost of all dense matrix multiplications in the transformer blocks (3 dense layers in FFN, 4 dense layers in the attention projections). Consequently, we can estimate the cost of the safety assessment with our for method, when using a shared covariance matrix as:

$$\text{FLOPS}_{\text{MLPM}} = 2 \cdot 1 \cdot \hat{N} \cdot d_{model} \cdot c, \tag{7}$$

where $\hat{N}$ refers to the number of layers we use for MLPM computation, and $c$ refers to the number of classes we distinguish between (in the simplest case, $c = 2$, as we only care about safe and unsafe distinction). Therefore, the upper bound on the ratio of the cost of the MLPM classification to the total prefill cost can be estimated as:

$$\frac{\text{FLOPS}_{\text{MLPM}}}{\text{FLOPS}_{\text{Prefill}}} = \frac{\hat{N} \cdot c}{s \cdot (2 \cdot v + N \cdot (3 \cdot d_{intermediate} + 4 \cdot d_{model}))}. \tag{8}$$

Assuming Llama3-8B model architecture, this ratio becomes:

$$\frac{\text{FLOPS}_{\text{MLPM}}}{\text{FLOPS}_{\text{Prefill}}} = \frac{\hat{N} \cdot c}{s \cdot (2 \cdot 128256 + 32 \cdot (3 \cdot 14336 + 4 \cdot 4096))} = \frac{\hat{N} \cdot c}{2157056 \cdot s}. \tag{9}$$

In practice, with $c$ small and $\hat{N}$ bounded by the total number of layers, the cost of safety assessment using our method during the generation phase is negligible.

**Memory Consumption**   When using a shared covariance matrix for GDA, the total number of parameters of MLPM is given by $\hat{N} \cdot (3d_{model}^2 + 1)$, which accounts for the $\hat{N}$ matrices $W = \Sigma^{-1}\mu$, biases $b = \mu^T\Sigma^{-1}\mu$, and aggregation weights ($w$) that our method stores. For a model with $d_{model} = 4096$, the single-layer params in half-precision take $\sim 24$KB when using separate covariance matrices. Assuming 32 layers, even if MLPM used all the layers for final detection, it would only need $\sim 768$KB, while guard models typically need $\sim 16$GB.

## M. Real Overhead Comparision

To assess the real-world efficiency of our approach, we profiled the end-to-end latency overhead introduced by our method. Specifically, we measured the MLPM execution time on Llama-8B for varying numbers of prototypes and compared these figures against baseline prefill times across multiple sequence lengths and batch sizes. The results, reported in milliseconds in Table 17, demonstrate that the MLPM overhead is negligible, typically well under 1 ms.

*Table 17.* End-to-end latency comparison (in ms) across several batch sizes between MLPM and standard LLM prefill for Llama-8B. We benchmark MLPM with all layers used for prototypes, 8 layers used for prototypes and just a single prototype. We also benchmark prefill times with varying sequence lengths ($L$). All execution times are reported as "mean $\pm$ standard deviation".

| Batch Size | MLPM-All | MLPM-8 | MLPM-1 | Prefill-$L$=256 | Prefill-$L$=512 | Prefill-$L$=1024 |
|---|---|---|---|---|---|---|
| 1 | $0.212_{\pm0.011}$ | $0.263_{\pm0.078}$ | $0.134_{\pm0.004}$ | $39.507_{\pm0.219}$ | $68.325_{\pm0.154}$ | $118.19_{\pm0.293}$ |
| 2 | $0.237_{\pm0.004}$ | $0.220_{\pm0.013}$ | $0.165_{\pm0.025}$ | $62.985_{\pm0.125}$ | $117.07_{\pm0.500}$ | $222.50_{\pm0.917}$ |
| 4 | $0.710_{\pm0.791}$ | $0.235_{\pm0.017}$ | $0.156_{\pm0.012}$ | $116.33_{\pm0.238}$ | $219.98_{\pm1.027}$ | $430.39_{\pm1.588}$ |
| 8 | $0.828_{\pm0.007}$ | $0.264_{\pm0.021}$ | $0.192_{\pm0.027}$ | $219.22_{\pm0.688}$ | $425.96_{\pm1.559}$ | $844.71_{\pm1.951}$ |

# N. MLPM with Hybrid Architectures

In this section, we provide additional results for Hybrid Architectures. We analyze how importance is distributed between the SSM and Attention layers in Table 18. SSM layers contribute most to our method's performance, and investigating this could be an interesting area for future work.

*Table 18.* Comparison of SSM and Transformer Attention layer utilization across Nemotron architectures. For each model, we show the fraction of selected SSM / ATN layers (the layers where the MLPM weight is non-zero). We also show their relative contributions to the total layer weights for the final MLPM aggregation, computed as the sum of the normalized MLPM weights across all SSM/ATN layers.

| Model | SSM Selected | SSM Weight sum | ATTN Selected | ATTN Weight sum |
|---|---|---|---|---|
| Nemotron-Nano-9B-V2 | 18/23 | 0.8181 | 3/4 | 0.1819 |
| Nemotron-3-Nano-4B | 15/15 | 0.7293 | 4/4 | 0.2707 |
| Nemotron-3-Nano-30B-A3B | 18/18 | 0.7678 | 6/6 | 0.2322 |

# O. MLPM vs Single Best Layers

Table 19 presents results for a single-layer GDA applied to the best-performing layer compared to MLPM. For all four models, MLPM outperforms the single-layer method. We also highlight that MLPM does not use any information from the evaluation set, whereas here the best layer is selected based on the evaluation performance.

*Table 19.* MLPM vs best layer

| Model | Best layer F1 | MLPM F1 | MLPM Gain |
|---|---|---|---|
| Mistral-7B-Inst | 86.98 | 87.55 | 0.57 |
| Llama-8B-Inst | 87.43 | 88.30 | 0.87 |
| OLMo2-7B-Inst | 89.78 | 90.23 | 0.45 |
| Qwen3-8B | 85.88 | 85.95 | 0.07 |

# P. LLM-as-a-Judge baseline

We evaluated our method against a naive prompting baseline (see Table 20). We calculated prompting performance using 10 distinct prompts that ask the model to assess the safety of the input and report both the average performance across all prompts for each sample and the maximum performance of the prompting baseline. For each dataset, we choose the best-case prompt.

*Table 20.* Comparison of prompting-based (LLM-as-a-Judge) baselines against MLPM.

| Model | Prompting(Avg) | Prompting(Max) | MLPM |
|---|---|---|---|
| Llama-8B | 75.20 | 82.75 | 87.55 |
| Mistral-7B | 72.28 | 85.63 | 88.30 |
| OLMo2-7B | 75.20 | 85.75 | 90.23 |

## Q. Multiclass Performance

To evaluate performance in a multiclass setting, we used the safety categories in WildGuardMix as labels, yielding a 15-class setup (1 safe, 14 unsafe). As shown in the table below, MLPM maintains high binarized accuracy ($> 81\%$) while achieving strong multiclass accuracy ($\sim 65\%$) across all tested models, demonstrating its applicability to the fine-grained classification tasks. The results are available in Table 21

*Table 21.* Performance on Multiclass safety assessment on WildGuardMix data of MLPM.

| Model | Binarized Acc | Multiclass Acc | AUROC |
|---|---|---|---|
| Llama-3B-8B | 0.8279 | 0.6617 | 0.8380 |
| Mistral-7B | 0.8141 | 0.6545 | 0.8345 |
| OLMo2-7B | 0.8494 | 0.6659 | 0.8336 |
| Qwen3-8B | 0.8153 | 0.6527 | 0.8350 |

## R. Development of Safety Representations During Training

To further investigate the effect of MLPM's backbone on performance, we evaluate our method with intermediate checkpoints from training OLMO2-7B and present the results in Figure 6; this analysis demonstrates how latent spaces and the features responsible for safety in LLMs evolve throughout training, and further quantifies our method's dependence on the backbone.

*Table 22.* OLMo2 Checkpoints

| Model | Num Tokens | Mean F1 |
|---|---|---|
| OLMo2-Stage1-Step150-Tokens1B | 1B | 50.13 |
| OLMo2-Stage1-Step600-Tokens3B | 3B | 52.30 |
| OLMo2_Stage1-Step1000-Tokens5B | 5B | 54.22 |
| OLMo2_Stage1-Step2000-Tokens9B | 9B | 58.55 |
| OLMo2_Stage1-Step3000-Tokens13B | 13B | 64.53 |
| OLMo2_Stage1-Step4000-Tokens17B | 17B | 66.38 |
| OLMo2_Stage1-Step5000-Tokens21B | 21B | 68.43 |
| OLMo2_Stage1-Step6000-Tokens26B | 26B | 69.97 |
| OLMo2-Stage1-Step10000-Tokens42B | 42B | 72.13 |
| OLMo2-Stage1-Step50000-Tokens210B | 210B | 76.90 |
| OLMo2-Stage1-Step200000-Tokens839B | 839B | 78.40 |
| OLMo2-Stage1-Step400000-Tokens1678B | 1680B | 79.21 |
| OLMo2-7B | 5000B | 79.99 |
| OLMo2-7B-Inst | 6000B | 90.16 |

