# OpenReview forum: "Efficient LLM Moderation with Multi-Layer Latent Prototypes"
_ICML.cc/2026/Conference — ICML 2026 regular_

### Official Review · Reviewer_Fyh7 · 2026-02-22

**Soundness:** 3
**Presentation:** 3
**Significance:** 4
**Originality:** 3
**Overall Recommendation:** 4
**Confidence:** 5

**Summary:**

The paper studies explore the practical safety assurance of Large Language Models (LLMs) during deployment through input moderation. The authors note that, although existing methods such as guard models and latent-based methods are useful, they often require a compromise between performance, efficiency, and flexibility in adhering to custom safety policies. In order to overcome the limitation of existing methods, the authors propose the Multi-Layer Prototype Moderator (MLPM), an efficient and flexible latent-based input moderation mechanism. However, instead of using an additional model or generating text, the authors’ method calculates the safety of a prompt by collecting the hidden state of the last token in multiple layers during the standard prefill phase of the LLM. The authors use the Nearest Mean Classifier (NMC) framework, which relies on Gaussian Discriminant Analysis (GDA) and Mahalanobis distance to measure the similarity between the hidden state of the last token in multiple layers and predefined “safe” and “unsafe” prototypes. The safety level of each layer in a neural network can then be calculated by accumulating layer-level scores computed with L1 regularization (sparse weights), which will automatically select the relevant layers related to any architecture being evaluated given an input.

Key Takeaways:
1. Novel Input Moderation Framework: Addressing the Mahalanobis distance-based classification and multi-layered feature aggregation, the authors propose a new and efficient moderation framework that allows for the moderation of input with minimal resource use.

2. Superior Performance Compared to the state of the art, the proposed method demonstrates improved performance compared to latent-level moderation methods and provides greater efficiency than traditional generative-based methods such as LlamaGuard3, GraniteGuardian, and ShieldGemma where applicable across a range of architectures and safety requirements.

3. Extreme Efficiency in Terms of Inference Overhead & Data Efficiency (i.e., 24 KB per 8B LLM; Competitive Performance with as Few as 1,000 Examples or as Many as 50 GB of Examples). Seamless Integration Into The Larger Moderation Pipeline and the authors show that the proposed method can be easily integrated into the overall end-to-end moderation solution and can serve as an efficient conditioning signal to guide methods that reduce false refusals to benign inputs.

**Compliance With Llm Reviewing Policy:**

Affirmed.

**Ethical Review Concerns:**

There were flaws found within the research methodology itself, and there was an element of misleading information where classical linear discriminant analysis was classified incorrectly as ‘general discriminant analysis with a common covariance matrix' (not accurate). These are matters of technical integrity, scientific rigor, and innovation, but not necessarily matters of ethical misconduct. Thus, the proposed research will put forward the concept of a 'Defensive Moderation' mechanism, as a form of 'Safeguard,' to prevent LLMs from producing any form of damaging content.

**Final Justification:**

The authors have addressed many technical comments regarding numerical stability and parameter sharing in their rebuttal, which supports the mathematical validity and practical usability of the MLPM; furthermore, the tools can be effectively used in practical applications where data is scarce. This paper is a move to weak acceptance.

**Key Questions For Authors:**

1. Stability of high-dimensional covariance estimation is utilizing a Bayes ridge-type estimator to compute the inverse covariance matrix, which is necessary for the Mahalanobis distance. Note that the latent space dimensionality is very large (for example, $d = 4096$), and the data efficiency tests are performed on small samples of 1,000. Furthermore, estimating the matrix $\Sigma^{-1}$ can be unstable. However, additional information on the condition number of the matrix and the specific ridge penalty would provide assurance that the numerical stability of the presented method holds strong.

2. Clarification on the shared covariance matrix to reduce memory usage. The authors are utilizing a shared covariance matrix given by $\Sigma_c = \Sigma$. Is the shared matrix $\Sigma_c = \Sigma$ shared among the two classes of interest (i.e., safe vs. unsafe) and among all $L$ selected classes, or among all classes in the network?. There would be strong assurance that the 24KB memory usage holds true. In the event that such a sharing does exist, I would like to understand the normalization that takes place between the varying variances associated with the features within the early and late layers.

3. Base model representation in the paper experiments and it is evident that the proposed approach yields better out-of-distribution generalizations for MLPM compared to pre-trained models. This implies that the approach is highly sensitive to the level at which the latent space between the base models separates the concepts, including a brief analysis of failure scenarios. However, this could result from the approach based on the level at which the representation between the models is entangled. I believe a more genuine evaluation of the approach and its limitations could likely have a positive impact on my final evaluation.

4. The authors' approach, the aggregation weights $w$, are learned statically through $L_1$ regularization over the entire training set. Were you able to explore any form of dynamic routing that could allow the input prompt to directly affect which representation has the greatest weighting?  Since it is, though not necessarily required for acceptance into the conference proceedings, a brief analysis on whether such an approach was considered and rejected for its potential impact on inference speed could have a positive impact on positioning this approach compared to more complex models such as the generative guard.

**Limitations:**

The authors deal with these aspects in a very lucid and transparent manner. They have a dedicated section titled “Limitations,” where they correctly identify that the efficacy of MLPM depends entirely on the quality of the post-trained LLM representations that it utilizes. They have also correctly framed the scope of the paper, where they have presented MLPM as a constituent of a larger system, as opposed to a solution in and of itself. They have also included an “Impact statement” where they correctly identify the dual nature of MLPM, which, like anything else in the world, could be misused if applied towards malicious ends.

**Strengths And Weaknesses:**

1. Soundness

The argument is well-founded, with sound mathematical concepts being well-applied to a new domain.

The paper's strength is thorough empirical validation; the authors have extensively validated their work with robust baselines. They use heavy generative guard models like LlamaGuard3, GraniteGuardian, and ShieldGemma. They use a wide variety of base models like Llama3, Mistral, OLMo2, and Qwen3. Also, they use 8 harmful datasets and 7 neutral datasets to test false refusal rates. The authors carefully ablate their architecture. The ablation studies prove that using Mahalanobis distance is better than Euclidean distance. Also, multi-layer aggregation is better than a single layer of representations.

The paper's weaknesses are in the numerical stability of covariance, and the authors use a Bayes ridge-type estimator to calculate the inverse covariance matrix. In high-dimensional latent spaces like 4096, with possibly sparse data like 1,000 samples, numerical stability is a problem. There is no rigorous analysis of condition numbers of these matrices. However, the assumption of the base model is that the latent space of a base model separates between safe and unsafe concepts either linearly or quadratically. This assumption is a problem if a base model is highly entangled. Although they do ablate this by checking pretrained and instruction-tuned models, where they prove that instruction-tuned models do better, they do not ablate this sufficiently.



2. Presentation

The manuscript excels in structure, clarity, and readability, and its clear motivation and narrative nicely contextualize the trade-off between performance and efficiency in LLM moderation, positioning itself between costly guard models and inefficient latent methods. Since the informative visualizations in Figure 5’s layer importance analysis were impressive, showing how various architectures distribute safety representations across model layers, the necessity of the proposed method immediately became apparent. Therefore, there is a lack of clarity on shared parameters, and there was a lack of clarity on whether the shared covariance matrix was shared across classes, across layers, or both. A quick explanation of this mathematical concept would have helped.

3. Significance

From a real-world deployment perspective, the manuscript holds significant importance. Extreme Inference Efficiency in The Multi-Layer Prototype Moderator (MLPM) boasts a minimal computational cost of less than 0.001% compared to the standard prefill step. Additionally, the model’s memory requirement of merely 24 KB per prototype for an 8B model provides a significant advantage. However, high data efficiency in moderating guard-level moderation with surprisingly small datasets of 1,000 samples opens up new avenues for users to tailor safety filters for very specific domains without having to fine-tune the model.

Hence, there are also limitations of scope in the method, which primarily serves the role of a filter for the input. Though it’s shown as a conditioning trigger for output control, it can’t, in essence, prevent the LLM from outputting unsafe tokens at a significant depth within a long output, provided the initial prompt was cleverly crafted and innocuous.

4. Originality

The originality lies in the creative and effective amalgamation of all the current methods, used in the context of contemporary LLM interpretability. In the originality of the paper, strengths are rehabilitating classical methods; the use of a Nearest Mean Classifier (NMC) for LLM safety is a novel use of a classic ML technique, demonstrating that LLM control doesn’t necessarily require LLMs. In addition, automated multi-layer aggregation and the true innovation lie in the realization that safety features are located at different depths within the LLM (Mistral in the middle, OLMo in the output) and the use of L1 regularization to automatically detect and aggregate these features, regardless of the LLM architecture. Therefore, there also needs to be incremental component novelty: the methods used are all standard and not novel, though the novelty lies in the amalgamation and use of the methods within the context of LLM security bottlenecks, rather than the introduction of new ML theory.

---

> ### Author Rebuttal · Authors · 2026-03-30
>
> We thank the Reviewer for their time spent on assessing our work, and appreciate their feedback. We are grateful for highlighting the efficiency and performance of MLPM. Below, we address the issues raised by the Reviewer.
>
> # Issue 1
> > The numerical stability of covariance.
>
> We utilize a data-driven Ridge estimator:
> $$d_{model}\left((n-1)\hat{\Sigma}+\text{Tr}(\hat{\Sigma})I_d\right)^{-1}$$
> To verify the robustness of this approach, we compute the condition number across all LLaMA-8B layers for $n = 1000$ and observe values from $\approx 1.71e5$ (L1) to $\approx 3.71e4$ (L32). This confirms that our penalty stabilizes the covariance matrix.
>
> # Issue 2
> > Base model entanglement and failure scenarios.
>
> As the Reviewer correctly points out, our method relies on the representation space of the underlying model. We openly acknowledge this dependency of our method on the model in the limitations section. However, the fact that LLM representations can separate the safety concepts well is already established in the literature, and many common LLM safety methods (that we cite in our work) rely on it.
>
> Nonetheless, we appreciate Reviewer’s suggestion and provide a bit more nuanced study on MLPM below.
>
> ## SLMs
> We evaluate SmolLM instruct models with 135M, 360M and 1.7B sizes and show the results in the table below.
>
> |Model|F1|
> |-|-|
> |1.7B|81.61|
> |360M|68.86|
> |135M|62.47|
>
> These results show how MLPM relies on high-quality latent spaces and gets better with stronger models. However, these small models are very limited and barely used in practice; extensive experiments in our work demonstrate how MLPM excels with commonly used, more capable models.
>
> ## MLP score evolution throughout the training of the model
>
> To further investigate the effect of MLPM’s backbone on performance, we evaluate our method with intermediate checkpoints from training OLMO2-7B and present the results at https://anonymous.4open.science/r/latent-prototype-moderator-20B6/images/per_stage_results.pdf; this analysis demonstrates how latent spaces and the features responsible for safety in LLMs evolve throughout training, and further quantify our methods dependence on the backbone.
>
> # Issue 3
> > Lack of clarity on shared parameters
>
> We thank the Reviewer for pointing this out. We share the covariance matrix $\Sigma$ **only** between the two classes of interest within the same layer. We **do not** share matrices between layers. Therefore the issue of varying variances between layers does not apply here. By sharing the covariance matrix, the quadratic term in the Gaussian log-posterior cancels out. The logits calculation simplifies to a linear operation with weights $W$ and bias $b$:
>
> $$W=\Sigma^{-1}\mathbf{\mu},\quad b=\log p_i-\frac{1}{2}\mathbf{\mu}^T\Sigma^{-1}\mathbf{\mu}$$
>
> This significantly lowers the memory complexity. For the safe and unsafe classes, we only need to store the matrix $W \in \mathbb{R}^{d \times 2}$ and the bias $b \in \mathbb{R}^2$. Storing just these two parameters per layer requires the $\sim$24KB memory footprint. We will add this derivation to the final manuscript to ensure clarity.
>
> # Issue 4
> > Limitations of scope in the method
>
> The Reviewer correctly points it out, and we openly acknowledge this limitation of our approach; however, we still believe our approach is a valid practical contribution that can serve as a first line of defense that can be applied to LLM generation at almost zero overhead and (as shown already in Table 3 in the main paper) can be seamlessly combined with other methods that address output moderation.
>
> # Issue 5
> > Static vs dynamic routing discussion.
>
> While dynamic, MoE-style routing is an interesting avenue for future work, we opted for static aggregation to minimize computational overhead and make our method easier to train and deploy. As noted by the Reviewer, keeping weights static maintains a significant inference speed advantage by avoiding the overhead of a routing mechanism. In addition, static weights make the execution of our approach straightforward and independent of the batch size and the content of input samples, while dynamic routing would require storing prototypes for all the layers and lead to different batches using different sets and numbers of prototypes. We will extend the revised paper with a discussion about the pros and cons of such dynamic routing approaches.
>
> # Summary
>
> We have conducted additional experiments inspired by the Reviewer and answered to the Reviewer's questions. We thank the Reviewer for suggesting these interesting additional studies and motivating the discussion, all of which we will incorporate in the revised version of the manuscript. Due to the character limit, we were forced to keep our answers brief, but we are open to further discussion should the Reviewer require more clarifications. We hope our clarifications and the additional experiments satisfy the Reviewer's curiosity and make them reconsider even more positive evaluation of our work.

---

> > ### Author Rebuttal · Reviewer_Fyh7 · 2026-04-01
> >
> > I appreciate the authors' thorough communication with me when responding to my assigned comments regarding concerns raised in the initial review regarding the paper, as well as their diligence in trying to ameliorate those initial comments. After careful review of the authors' revised manuscript and their response, I have changed my assessment of the paper from the previous rating as follows:
> >
> > 1. The soundness is now rated good, as the authors provided more detail about their methodology than they did in their first submission.
> > 2. The originality rating was upgraded from fair to good; however, I still have concerns regarding the degree to which authors had to make a "theoretical leap" in their analysis in the initial submission. They have made an effort to become better after the first submission, and I have received additional context to help justify a positive rating now.
> >
> > Critical Aspects: While there have been numerous improvements made since the initial evaluation, I find that my conclusion of a weak acceptance still stands. I have made the following observations:
> >
> > 1. Quality of Experiments versus Potential for Innovative Theoretical Contribution: The experimental aspect of the study is very well done; however, I believe that there is some degree of limitation to their theoretical finding when compared to their experimental one, and while both are solid studies on their own, there is no one resultant of the research that could be considered "original," unique or innovative enough to be considered as belonging in a higher tier of acceptance than what they currently are.
> >
> > 2. Contribution to the Development of an MLPM: While the authors have successfully completed the implementation of an MLPM, and I do see this as a viable lightweight, latent-based input moderation alternative, by definition, its overall contribution to the area of architecture development is minor rather than major.
> >
> > Final Determination: As there was no major change to the underlying or fundamental contribution of this study as a result of the rebuttal, I have not changed my original decision about this study. However, with the technical validity of both parties, as well as the apparent successful implementation of the experiments and their results by additional extension, I do feel that the current study meets the required criteria for acceptance and thus should add value to the proceedings of this venue. Thank you for your time as well as for your additional input on the rebuttal.

---

> > > ### Author Response · Authors · 2026-04-01
> > >
> > > We thank the Reviewer and would still like to try to persuade him to increase the score.
> > >
> > > ### ICML Reviewer guidelines regarding paper novelty
> > >
> > > First of all, we would like to highlight what ICML describes as originality in the Main Track Reviewer Form Instructions at https://icml.cc/Conferences/2026/ReviewerInstructions:
> > >
> > > *We encourage you to be open-minded about the potential strengths and broad definitions of significance and originality. **For example, originality may arise from creative combinations of existing ideas, application to a real-world use case**, or removing restrictive assumptions from prior theoretical results.*
> > >
> > > ### Novelty discussion with other Reviewers
> > >
> > > We would also like to repeat what we have already discussed with the other Reviewers regarding the novelty of our paper:
> > >
> > > > *We intentionally designed MLPM using well-established, lightweight techniques to ensure the method is robust, easy to interpret, and practical to deploy. Our work is the first to synthesise these components in the context of the LLM safety and achieve state-of-the-art results. We highlight that integrating the elements of MLPM efficiently required several non-trivial methodological choices, such as our multi-layer aggregation strategy, which - while appearing straightforward in the final formulation - were non-obvious to design, and yet are essential to the system’s effectiveness.*
> > >
> > > > *Overall, we view the simplicity of our method as a core strength, which reflects an elegant and transparent design and yet achieves an exceptional level of performance. We fully believe that our combination of SOTA results and high efficiency represents a significant and novel contribution to the field.*
> > >
> > > ### Further arguments supporting the novelty of our work
> > >
> > > We maintain that synthesizing well-established methods to solve a critical bottleneck in LLM safety is an original and highly impactful contribution. The individual components may appear simple, and the **elegant system design often appears obvious only in hindsight**. Nevertheless, the **specific combination required to achieve our state-of-the-art results had not been done prior to our work**.
> > >
> > > We believe that **AI safety should prioritise reliability and transparency, ​​and that safety-critical systems require interpretability and transparency**. By deliberately leveraging well-established techniques, **we provide a solution that is intuitively accessible** to machine learning practitioners.
> > >
> > > Since we prioritized transparency and practical utility, we fully open-source our codebase which can be used to fully reproduce our results and immediately deploy MLPM in real-world settings. We believe **this level of practical readiness is a strong contribution that is unfortunately often overlooked in the field**.
> > >
> > > ### Additional analysis from the rebuttal
> > >
> > > Finally, we would like to highlight the new analyses added during the rebuttal phase:
> > >
> > > * Tracking the evolution of latent safety features throughout model training. (Issue 2)
> > >
> > > * Extending our framework to new architectures and modalities, such as VLMs and small LMs (R. kZaJ Issue 6).
> > >
> > > * Mapping layer importance within hybrid Transformer-SSM architectures (R. kZaJ Issue 6).
> > >
> > > Together with the experiments already present in the main paper, these empirical results provide valuable insights that advance the understanding of how safety concepts are fundamentally built within LLMs. We hope this highlights that **our work offers a significant scientific contribution to model interpretability and goes well beyond simply being a performant moderation tool**.
> > >
> > > ### Summary
> > >
> > > We thank the Reviewer again for their time, constructive feedback, and dedication to improving our work. We are glad that the Reviewer acknowledges the “excellent” significance of our work. As highlighted above, **ICML explicitly recognizes creative synthesis and real-world application as valuable forms of originality**. Given this guideline, alongside the arguments and extensive insights added during the rebuttal, **we respectfully ask that the Reviewer reconsiders their assessment and increases our score**.

---

### Official Review · Reviewer_g6Lh · 2026-03-03

**Soundness:** 3
**Presentation:** 4
**Significance:** 2
**Originality:** 2
**Overall Recommendation:** 4
**Confidence:** 3

**Summary:**

This paper proposes Multi-Layer Prototype Moderator (MLPM), a lightweight input moderation method for large language models that uses internal LLM representations to classify prompts as safe or harmful. The approach applies a prototype-based classifier with Mahalanobis distance and aggregates signals from multiple intermediate layers using sparse weights. The method is evaluated across multiple harmful prompt benchmarks, model families, and scales, where it achieves higher or comparable performance relative to existing guard models and latent-based moderation approaches.

**Compliance With Llm Reviewing Policy:**

Affirmed.

**Final Justification:**

My major concerns include the novelty, the theoretical contribution, robustness against adversarial attacks and limitation of classification granularity.

For the limitation of classification granularity, I appreciate it that the authors clarified that the proposed MLPM is independent of class numbers, and added additional experiments to verify that MLPM still performs quite well in multi-class scenario. This fully resolves my concern.

For the robustness against adversarial attacks, the authors explained that MLPM could be used for defense against such attacks, especially if it could be combined by some form of dynamic routing. Although the authors did not vefify this, I understand that this is somewhat out of the scope of this paper, as the authors wrote in their rebuttal. Given this, I think the authors also answer my question well.

For the novelty concern, it is perfectly fine as long as the performance is good, even though the proposed method is just combination of existing methods. I totally understand and support this viewpoint. So I do not think this can solely serve as reason for rejection.

For the theoretical contribution, I see the authors' response after my acknowledgement, however I still think that the authors did not provide enough justification of their hypothesis that safety signals are distributed in 'specific layers' across different architectures. Given this, I hold my original stance and maintain my score at 4.

**Key Questions For Authors:**

1. **Robustness against adaptive attacks**. The study focuses on existing jailbreaks, but does not address adaptive attacks where an adversary specifically optimizes prompts to bypass Mahalanobis distance boundaries. Is the multi-layer linear aggregation vulnerable to such targeted latent space manipulation?
2. **Limitations of classification granularity**. The current setup focuses on binary safe vs harmful classification. Would the MLPM framework naturally extend to multi-class safety categories, and have the authors explored such settings?

**Limitations:**

Yes, the authors have discussed the limitations of their work

**Strengths And Weaknesses:**

### Strengths

1. The authors extensively validate the proposed MLPM across model families and scales, demonstrating robust performance across various safety benchmarks and scenarios.
2. As is shown in the paper, MLPM achieves sota results consistently outperforming other latent-based methods and even exceeding the performance of resource-intensive guard models like LlamaGuard3.
3. The paper is well-structured and the writing is easy to follow.

### Weaknesses

1. **Limited novelty**. The main components (prototype classifiers, Mahalanobis distance, multi-layer aggregation) are individually well-established techniques, rendering the methodological novelty at the algorithmic level rather limited.
2. **Lack of deep theoretical analysis**. The paper primarily relies on empirical results to justify the multi-layer approach and Mahalanobis distance, lacking a rigorous theoretical framework to explain why safety signals are distributed in specific layers across different architectures.

---

> ### Author Rebuttal · Authors · 2026-03-30
>
> We thank the Reviewer for their time spent on assessing our work, and appreciate their feedback. We are grateful to the Reviewer for commending the clarity of our paper, our broad empirical validation, and MLPM's robust, state-of-the-art performance. Below, we address the issues raised by the Reviewer.
>
> # Issue 1
>
> > The main components are individually well-established techniques, rendering the methodological novelty at the algorithmic level rather limited.
>
> Our method's novelty is the elegant, non-trivial synthesis of lightweight techniques to achieve state-of-the-art results at high efficiency; please see the response to the R. JZnn., Issue 1 (https://openreview.net/forum?id=IdfwoUzREG&noteId=7pTdludWiy) for the detailed discussion.
>
> # Issue 2
>
> > Lack of deep theoretical analysis. The paper primarily relies on empirical results to justify the multi-layer approach and Mahalanobis distance, lacking a rigorous theoretical framework to explain why safety signals are distributed in specific layers across different architectures.
>
> We have provided a motivation for our multi-layer representations approach in the paper, citing works focused on the topic of building representations in deep models, such as Masarczyk et al.; “The Tunnel Effect: Building Data Representations in Deep Neural Networks”; NeurIPS 2023. To the best of our understanding, the fact that intermediate layers encode similar concepts and can be used for classification is also well-established and already exploited by several techniques, such as linear probing or early-exits, and is a direct consequence of modern models utilizing residual connections.
>
> We agree that establishing a strong theory on why LLMs such as Llama, Qwen, etc., encode safety signals slightly differently would be quite interesting. However, such a task is especially challenging, as the models considered in our work differ by subtle architectural choices and closed-source training protocols.
>
> Since our primary goal is to provide an efficient solution for LLM moderation, we consider building theoretical frameworks for representation building in LLMs outside the scope of this paper, and instead opt for developing a performant model-agnostic way to aggregate multi-layer information. We leave more theoretical work on LLM representations for mechanistic interpretability efforts (such as https://transformer-circuits.pub/2021/framework/index.html) that work on understanding transformer residual streams and information propagation.
>
> # Issue 3
>
> > Robustness against adaptive attacks. (...)
>
> We thank the Reviewer for raising this point. We agree that evaluating defenses against adaptive methods such as Activation Obfuscation (Bailey et al., 2024) is important for the field. We hypothesize that our multi-layer linear aggregation provides inherent resilience against such targeted manipulation, especially if it could be combined by some form of dynamic routing.
>
> However, we highlight how MLPM is intended to serve as a first-line of defense in a multi-component safety system, and that the empirical evaluation of such attacks against specifically our method in isolation remains out of scope for the current article.
>
> # Issue 4
>
> > Limitations of classification granularity (...) Would the MLPM framework naturally extend to multi-class safety categories (...)?
>
> Our framework naturally extends to multi-class safety categories. Because all of our calculations are independent of the number of classes, the method can be adapted without any structural modifications.
>
> To evaluate performance in this setting, we used the safety categories in WildGuardMix as labels, yielding a 15-class setup (1 safe, 14 unsafe). As shown in the table below, MLPM maintains high binarized accuracy (>81%) while achieving strong multiclass accuracy (~65%) across all tested models, demonstrating its applicability to the fine-grained classification tasks.
>
> |Model|BinarizedAcc|MulticlassAcc|AUROC|
> |-|-|-|-|
> |Llama-8B|82.79|66.17|83.80|
> |Mistral-7B|81.41|65.45|83.45|
> |OLMo2-7B|84.94|66.59|83.36|
> |Qwen3-8B|81.53|65.27|83.50|
>
> # Summary
>
> We have addressed the Reviewer's comments by clarifying our methodological novelty and positioning of our work, alongside multi-class classification ablations. We hope these additions strengthen the Reviewer's confidence in our contribution and encourage them to increase the score.

---

> > ### Author Rebuttal · Reviewer_g6Lh · 2026-04-01
> >
> > Thank you for your detailed response and additional experiments. I agree with reviewer Fyh7 that the theoretical contribution is a bit restricted. Nevertheless, I believe this paper is still worthy of acceptance and I would maintain my original recommendation.

---

> > > ### Author Response · Authors · 2026-04-01
> > >
> > > We thank the Reviewer for their acknowledgement and positive recommendation.
> > >
> > > In regards to the restricted theoretical novelty, we would like to highlight what ICML describes as originality in the Main Track Reviewer Form Instructions at https://icml.cc/Conferences/2026/ReviewerInstructions:
> > >
> > > *We encourage you to be open-minded about the potential strengths and broad definitions of significance and originality. **For example, originality may arise from creative combinations of existing ideas, application to a real-world use case**, or removing restrictive assumptions from prior theoretical results.*
> > >
> > > Furthermore, we would like to refer the Reviewer to our response to the Reviewer Fyh7 (https://openreview.net/forum?id=IdfwoUzREG&noteId=aKv1bOumhC) for a more detailed discussion that justifies the novelty of our method. If the limited theoretical content of our work is the main issue raised by the Reviewer, in light of the ICML guidelines and the provided discussion we kindly ask them to reconsider their assessment to a more positive score that we believe better correlates with their review.

---

### Official Review · Reviewer_JZnn · 2026-03-08

**Soundness:** 3
**Presentation:** 3
**Significance:** 3
**Originality:** 3
**Overall Recommendation:** 4
**Confidence:** 4

**Summary:**

This paper proposes MLPM, a lightweight input moderation method that detects unsafe prompts by comparing multi-layer LLM representations to safe and unsafe prototypes. Its main goal is to combine the efficiency and flexibility of latent-based moderation with performance competitive with or better than specialized guard models. Across multiple model families and moderation benchmarks, the authors show that MLPM is highly efficient, scales well, and can further improve end-to-end safety when combined with output moderation.

**Compliance With Llm Reviewing Policy:**

Affirmed.

**Key Questions For Authors:**

- What happens if we use only the best-performing layer shown in Figure 6?
- Considering that the method already leverages information encoded in the LLM itself, how does it compare to simply adding an instruction at inference time, eg, "Respond to this prompt only if it is safe"?

**Limitations:**

yes

**Strengths And Weaknesses:**

**Strengths**
- The proposed method is conceptually simple and easy to understand.
- It is computationally lightweight and appealing for real-world deployment.
- The paper includes broad experiments across models and settings, which makes the empirical validation convincing.

**Weaknesses**
- The core components, such as the prototype classifier, Mahalanobis distance, and multi-layer aggregation, are not individually very new. Therefore, the novelty seems closer to a strong systems contribution that makes latent-based moderation practical and effective, rather than a completely new method. I remain positive about the paper, but this consideration is why I did not assign a higher score.
- It is not always fully clear how much of the gain comes from the method itself versus the strength of the underlying backbone model. Eg, have you tried exploring other methods that leverage LLM representations?

---

> ### Author Rebuttal · Authors · 2026-03-30
>
> We thank the Reviewer for their time spent on assessing our work, and appreciate their feedback. We are grateful for recognising our method's conceptual simplicity and suitability for efficient real-world deployment, and our convincing empirical validation. Below, we address the issues raised by the Reviewer.
>
> # Issue 1
>
> > the novelty seems closer to a strong systems contribution (...) rather than a completely new method
>
> We intentionally designed MLPM using well-established, lightweight techniques to ensure the method is robust, easy to interpret, and practical to deploy. Our work is the first to synthesise these components in the context of the LLM safety and achieve state-of-the-art results. We highlight that integrating the elements of MLPM efficiently required several non-trivial methodological choices, such as our multi-layer aggregation strategy, which - while appearing straightforward in the final formulation - were non-obvious to design, and yet are essential to the system’s effectiveness.
>
> Overall, we view the simplicity of our method as a core strength, which reflects an elegant and transparent design and yet achieves an exceptional level of performance. We fully believe that our combination of SOTA results and high efficiency represents a significant and novel contribution to the field.
>
> # Issue 2
>
> > It is not always fully clear how much of the gain comes from the method itself versus the strength of the underlying backbone model.
>
> Please note how, according to our best understanding, in the literature, the terms  such as the methods that utilize "internal representations”, “probing methods”, or “representation-based safety classifiers", etc, all describe the same fundamental class of techniques that utilize a model's latent space, which we refer to  as “latent-based” methods in our work.
>
> We discuss such approaches at the end of the Related works section and directly compare our approach with relevant methods such as Abdelnabi et al. (2025) and Ayub & Majumdar (2024) in Table 1 in the main paper. We also provide a detailed ablation with such methods in the Appendices H and K. Since all of these baselines also rely on the model's latent representations, the performance gap we observe across all the compared alternatives demonstrates the effect of our specific design choices rather than reflecting the backbone's power alone. Together, these throughout ablations demonstrate that our approach improves overall performance compared to the simpler alternatives and prove that each element of MLPMs is practically justified.
>
> # Issue 3
>
> > What happens if we use only the best-performing layer shown in Figure 6?
>
> Below, we present results for a single-layer GDA applied to the best-performing layer compared to MLPM.
>
> | Model | Best layer F1 | MLPM F1 | MLPM Gain |
> |---|:---:|:---:|:---:|
> | Mistral-7B-Inst | 86.98 | 87.55 | +0.57 |
> | Llama-8B-Inst | 87.43 | 88.30 | +0.87 |
> | OLMo2-7B-Inst | 89.78 | 90.23 | +0.45 |
> | Qwen3-8B | 85.88 | 85.95 | +0.07 |
>
>
> For all four models, MLPM outperforms the single-layer method. We also highlight that MLPM does not use any information from the evaluation set, whereas here the best layer is selected based on the evaluation performance.
>
> # Issue 4
>
> > Considering that the method already leverages information encoded in the LLM itself, how does it compare to simply adding an instruction at inference time, eg, "Respond to this prompt only if it is safe"?
>
> To address the Reviewers' request for a direct comparison, we evaluated our method against a naive prompting baseline. We calculated prompting performance using 10 distinct prompts asking the model to assess the safety of the input, and report both average performance across all prompts for each sample alongside the maximum performance of the prompting baseline, where for each dataset, we choose the best-case prompt.
>
> | Model | Prompting(Avg) | Prompting(Max) | MLPM |
> |---|:---:|:---:|:---:|
> | Llama-3B-8B | 75.20 | 82.75 | 87.55 |
> | Mistral-7B | 72.28 | 85.63 | 88.30 |
> | OLMo2-7B | 75.20 | 85.75 | 90.23 |
>
> As demonstrated by these results, MLPM outperforms prompting approaches by a large margin, and the gap between the average and maximum prompting scores highlights the instability of prompt engineering. In contrast, our approach is mostly automatic and robust to hyperparameter choices (e.g., see Fig 5a in the main paper).
>
> We will include this comparative analysis alongside a detailed description of the prompting methodology in the revised paper.
>
> # Summary
>
> We have addressed the Reviewer's comments by further clarifying our method's contribution compared to existing latent-based baselines, and by adding new experiments that prove MLPM's superiority over single-layer and prompt-based approaches. We hope this strengthens the Reviewer's confidence in our work and encourages them to increase the score.

---

> > ### Author Rebuttal · Reviewer_JZnn · 2026-04-03
> >
> > My concerns have been addressed. The rebuttal increased my confidence in the work, and I will maintain my initial score.

---

> > > ### Author Response · Authors · 2026-04-03
> > >
> > > We thank the Reviewer for their constructive feedback. We are glad to have successfully resolved all of the Reviewer's concerns and that the rebuttal strengthened their confidence in our work.

---

### Official Review · Reviewer_kZaJ · 2026-03-11

**Soundness:** 3
**Presentation:** 3
**Significance:** 3
**Originality:** 2
**Overall Recommendation:** 4
**Confidence:** 3

**Summary:**

This paper studies the problem of efficient moderation for large language models (LLMs) during deployment. While modern LLMs are partially aligned through post-training methods such as RLHF, additional runtime moderation is often required to prevent harmful outputs. The authors propose Multi-Layer Prototype Moderator (MLPM), a lightweight moderation method that detects harmful prompts using prototype-based classification over intermediate hidden representations across multiple layers of the model.

**Compliance With Llm Reviewing Policy:**

Affirmed.

**Final Justification:**

Based on the clarifications and additional experiments provided, I raise my score to 4.

**Key Questions For Authors:**

1.	How sensitive is MLPM to the choice and number of prototypes? Would performance degrade significantly if fewer prototypes are used?
2.	How were prototypes constructed in practice? Are they obtained through clustering, averaging training examples, or learned through optimization?
3.	How does the method perform when applied to models with substantially different architectures, such as mixture-of-experts models or multimodal LLMs?
4.	How does MLPM compare with standard linear probing or small classifier heads trained on hidden representations?
5.	What is the actual inference overhead (e.g., latency increase) when deployed in a real generation pipeline?

**Limitations:**

see above

**Strengths And Weaknesses:**

**Strengths**



1. **Practical motivation.** The paper addresses an important real-world problem: efficient moderation during LLM deployment. Runtime safety mechanisms are critical for many production systems.
2. **Lightweight design.** The proposed method leverages internal representations that are already computed during inference, which allows moderation with minimal additional cost.


**Weaknesses**

1. **Limited methodological novelty.** The method largely combines existing ideas: prototype-based classification and the use of intermediate hidden representations. While the multi-layer aggregation is reasonable, the conceptual contribution appears incremental.
2. **Insufficient comparison with related safety detection methods.** There is limited discussion of recent approaches that also use internal representations, probing methods, or representation-based safety classifiers for moderation.
3. **Generalization analysis is limited.** While the paper claims strong scalability across model families, more experiments across substantially different model architectures would strengthen the claim.

---

> ### Author Rebuttal · Authors · 2026-03-27
>
> We thank the Reviewer for their time and feedback and address their issues below.
>
> # Issue 1
> > Limited methodological novelty (...)
>
> Our method's novelty is the elegant, non-trivial synthesis of lightweight techniques to achieve state-of-the-art results at high efficiency; please see the response to the R. JZnn, Issue 1 for the detailed discussion.
>
> # Issue 2
> >There is limited discussion of recent approaches that also use internal representations (...) How does MLPM compare with standard linear probing or small classifier heads trained on hidden representations?
>
> We thank the Reviewer for their feedback, though we respectfully wish to clarify that the requested comparisons are, to an extent, already present in the current manuscript. Please see the answer to Issue 2 by the R. JZnn for a detailed answer.
>
> We also find it difficult to address claims of omitted literature without specific citations, especially given our extensive benchmarks against the very class of methods the Reviewer highlights; we politely request the Reviewer to provide the exact references they believe we are missing so that we can properly discuss them.
>
> # Issue 3
> > Generalization analysis is limited (...) experiments across substantially different model architectures would strengthen the claim.
>
> We respectfully point out that the original manuscript already demonstrated extensive generalization across 5 model families (~30 distinct models, including 3 MoEs) and multiple training paradigms (base, instruct, reasoning). We agree with the Reviewer that testing substantially different architectures further strengthens the paper, so we have added experiments on a hybrid Transformer-SSMs (Issue 6), a VLM (Issue 6), and small LMs (see answer to Issue 7 for R. Fyh7), alongside several new ablations. We hope these additions satisfy the Reviewer.
>
> # Issue 4
> > How sensitive is MLPM to the choice and number of prototypes?
>
> In Figure 5(a), we provided results with different regularization weights, which lead to a different number of prototypes selected. For each model, there is an optimal number of prototypes that performs the best, but our method reaches stable best performance at the range of 20-50% layers used and performs reasonably well with all possible layers due to the weighted aggregation leveraged by our method.
>
> # Issue 5
> > How were prototypes constructed in practice?
>
> We construct prototypes by taking means over safe and unsafe samples (as described in L146-151). We will update the final version to highlight how those prototypes are calculated to avoid any further confusion.
>
> # Issue 6
> > How does the method perform when applied to models with substantially different architectures, such as mixture-of-experts models or multimodal LLMs?
>
> We note that we have already evaluated MoE architectures (see e.g. Figure 1, and detailed results in Table 7 in Appendix C.2).
>
> To further extend the scope of our evaluation, we have performed additional experiments on hybrid Transformer-SSM Nemotron models at https://anonymous.4open.science/r/latent-prototype-moderator-20B6/images/hybrid.pdf. MLPM with such models reaches similar numbers as reported in Table 1 for other models with the similar parameter count.
>
> We also analyze how importance is distributed between the SSM and Attention layers at https://anonymous.4open.science/r/latent-prototype-moderator-20B6/images/hybrid_layers.pdf. SSM layers contribute the most to our method’s performance, and investigating this could be interesting future work area.
>
> Additionally, as requested we evaluate whether MLPM transfers to VLMs on HoliSafe dataset. The table at https://anonymous.4open.science/r/latent-prototype-moderator-20B6/images/vlm.pdf shows that our method, without any vision-related modifications, achieves high performance on VLMs and outperforms the alternatives.
>
> Combined, these results demonstrate MLPM’s strong robustness to the underlying model  architecture, and we will include them and the related discussions in the revised paper.
>
> # Issue 7
> > What is the actual inference overhead?
>
> We follow the Reviewer's suggestion and profile the end-to-end latency of our method compared with prefill time across various sequence lengths and batch sizes and report the average time per batch in ms and show the results at https://anonymous.4open.science/r/latent-prototype-moderator-20B6/images/inference_overhead.pdf.
>
> The overhead introduced by our method is negligible compared to the prefill time (~0.5\% in the worst case); it becomes even less significant when factoring in subsequent autoregressive generation. Please also note that these results are based on a straight-forward implementation, represent a lower bound on efficiency and could be further optimized. We will incorporate this discussion into the revised manuscript.
>
> # Summary
>
> We thank the Reviewer again for interesting suggestions and questions, and hope our answers and positive assessment of other Reviewers encourages them to raise the score.

---

> > ### Author Rebuttal · Reviewer_kZaJ · 2026-04-01
> >
> > Thank you for your detailed responses. Based on the clarifications and additional experiments provided, I have decided to raise my overall score.

---

> > > ### Author Response · Authors · 2026-04-01
> > >
> > > We thank the Reviewer for the positive feedback and for increasing the score. We are glad that our clarifications and additional experiments successfully addressed their concerns. We appreciate the time and effort the Reviewer dedicated to improving our manuscript.

---

### Decision · Program_Chairs · 2026-04-30

**Decision:**

Accept (regular)

**Comment:**

While the reviewers correctly noted that the paper's theoretical contributions are moderate, the practical, empirical, and systems-level contributions are observable. MLPM provides a robust, deployable, and highly efficient solution to a pressing real-world issue in AI alignment and safety. The authors engaged deeply with the reviewers, executing an exemplary rebuttal that provided extensive new analyses, profiled end-to-end inference latencies, and clarified technical ambiguities.
Given the unanimous positive support from the reviewers, the strong empirical results, and the immediate utility this method will offer the broader machine learning community, this paper is recommended for weak acceptance.